# Admixed and single-continental genome segments of the same ancestry have distinct linkage disequilibrium patterns

Hanbin Lee[1,2,3*†], Moo Hyuk Lee[2,4†], Kangcheng Hou[5], Bogdan Pasaniuc[6,7] and Buhm Han[2,4,8,9,10*]

†Hanbin Lee and Moo Hyuk Lee contributed equally to this work.

*Correspondence:
hblee@umich.edu; buhm.han@snu.ac.kr

[1] Present Address: Department of Statistics, University of Michigan, Ann Arbor, MI, USA
[4] Department of Biomedical Sciences, Seoul National University College of Medicine, Seoul, Republic of Korea
Full list of author information is available at the end of the article

## Abstract

**Background:** Admixed populations offer valuable insight into the genetic architecture of complex traits. Many studies have proposed methods for genome-wide association study (GWAS) in admixed populations and various simulation studies have evaluated their performances. In this work, we propose another direction of comparison of recently proposed methods for admixed GWAS from a population genetic viewpoint.

**Results:** Our theoretical approach mathematically and directly compares the power of methods given that the causal variant is tested. This is done by deriving the variance formula of the methods from the population genetic admixture model. Our results analytically confirm previous observation that the standard GWAS test is more powerful than alternative tests due to leveraging allele frequency heterogeneity in which alternatives do not. As a by-product, we obtain a simple method to improve the power of multi-degrees-of-freedom tests only using summary statistics. We further investigate the problem when the causal variant is not directly known but is detected by tagging variants in linkage disequilibrium (LD). The analysis shows that a genetic segment from admixed genomes may exhibit distinct LD patterns from the single-continental counterpart of the same ancestry.

**Conclusions:** While the classic admixture model is successful in predicting GWAS power, its popular extension in the literature falls short in explaining the LD patterns found in simulations and real data, warranting an improved model for LD in admixed genomes.

**Keywords:** Genome-wide association study, Admixture, Population genetics

## Background

The Pritchard-Stephens-Donnelly (PSD) model, or the STRUCTURE, has been widely used to infer the population structure of admixed populations [1–6]. In this model, population structure is a latent variable called ancestral proportion (AP), also called global ancestry. Allele frequencies are then represented as a weighted average of

ancestry-specific allele frequencies, where the ancestral proportions are the weights. When inferring the ancestral proportion, it is assumed that the loci used in the analysis are approximately independent without linkage disequilibrium. Contemporary genomes with ancestry labels are used as surrogates of the ancestral genomes to determine the ancestry-specific frequencies. A related variable is the local ancestry (LA) which is the source population label of chromosomal segments. Methods predicting LA view admixed genomes as a mosaic of single continental genomes [7–10]. These methods reflect the underlying evolutionary process more faithfully than the PSD model by incorporating the effect of genetic drift and recombination [11, 12].

Both global and local ancestries are used extensively in complex trait analysis of admixed populations. Global ancestry adjustment is an essential ingredient of GWAS to control population structure [13, 14]. Although principal components (PCs) are generally used instead, PCs can be expressed as a linear combination of ancestral proportions under many demographic processes, making the two adjustments equivalent [15, 16]. Some choose to include local ancestry as covariates to handle the fine-scale structure of admixed genomes [17–20].

The PSD model was originally designed to infer population structure and not for GWAS per se. This is because GWAS relies on LD which is a two-loci property, while the PSD model describes the marginal distribution of a single locus [1, 21]. Nevertheless, a simple extension to incorporate the two-loci distribution is found in the literature. Although implicit in many cases, the extension assumes that the length of the local ancestry segment is far longer than the range of within-continental LD [22–24]. This makes the segment's local LD structure the same as the source population of a single continental origin. Technically speaking, it means that variants on different local ancestry segments are independent given the local ancestry. Examples include the simulation in the original Tractor paper and several polygenic score methods tailored for admixed populations [20, 22, 23]. Tractor produces ancestry-specific estimates and it does not necessarily require the extended model to hold. However, the simulation did not take LD into account and assumed that the tested variant is causal without being in LD with other variants, which falls into the case in which the extended model holds. We will call this extension of the PSD model, which stipulates the within-continental LD to be far shorter than the length of local ancestry segments, as the extended PSD (ePSD) model.

In this work, we show that PSD and ePSD models can generate many useful predictions that can be empirically verified in real data. The PSD model, in particular, provides closed-form formulas for the standard error of two popular GWAS methods: the standard GWAS (or the Armitage trend test, ATT) and Tractor. With the additional assumption that the tested variant is a causal variant not in LD with other variants, we find that the power advantage of standard GWAS over Tractor is proportional to the allele frequency difference between the source populations. While deriving the formula, we found that Tractor estimates of the ancestry-specific effects are independent despite coming from the same set of individuals. This allows us to combine Tractor estimates with existing summary statistics meta-analysis tools to improve power. Nevertheless, this strategy still cannot achieve the power of standard GWAS as we show by theory and in real data if we test the causal variant.

We extend our analysis to the case of imperfect tagged causal variants. In this case, we test variants that are in LD with unobserved causal variants. The PSD model is not enough in this case, so we adopt the ePSD model to handle linkage disequilibrium between observed markers and unobserved causal variants. Our theory clarifies the role of global and local ancestries for correcting genetic confounding in GWAS. Genetic confounding occurs when a causal variant that is far from the tested variant is omitted in the regression [21, 25]. As long as we adjust global ancestry, in both standard GWAS and Tractor, effect sizes are a linear combination of casual variants tagged by the tagging SNP weighted by LD correlations. Local ancestry only absorbs the residual effects of these tagged causal variants that are not perfectly captured by the tagging variant. Hence, local ancestry does not capture the effect of distant causal variants not included in the regression, not contributing to genetic confounding correction.

We indirectly tested ePSD empirically based on its prediction. We found that under the ePSD model, Tractor estimates are not only unbiased if the tested variant is the causal variant (as shown in the original paper's simulation), but also unbiased when it is a marker variant that potentially tags multiple causal variants. This means that Tractor estimates can be supplied to linkage disequilibrium score regression (LDSC) using the single-continental reference panel. Using the summary statistics produced from admixed genomes of the Population Architecture using Genomics and Epidemiology (PAGE) cohort and single-continental summary statistics from the Pan UK Biobank (PanUKBB), we attempted to compute the genetic correlation between ancestry-specific Tractor estimates and single-continental GWAS summary statistics from African and European GWAS of 19 quantitative traits. The genetic correlation estimates were generally below 1, suggesting that the prediction produced by ePSD does not hold well. However, the wide confidence intervals prevented us from drawing strong conclusions.

To more directly address the issue, we simulated 10,000 admixed genomes and compared their LD patterns to 10,000 genomes of single-continental origin. We compared the same-ancestry segments from admixed and single continental genomes, unlike the previous study that compared African and European segments to each other [26]. The concordance between Tractor's ancestry-specific estimates from admixed genomes and effect sizes from single-continental genomes was high when the polygenicity was low. The concordance dropped quickly as the polygenicity grew, which led to an increased number of causal variants tagged by the marker variant. We found that the low concordance is driven by variants that are relatively distant from the marker variant. The LD of these variants with the marker variant in admixed genomes differed from that in their single-continental counterparts, disqualifying the ePSD model's key assumption that within-continental LD cannot stretch beyond local ancestry segments.

## Results

### A non-technical overview

This paper deploys the Pritchard-Stephens-Donnelly (PSD) model and its extension (ePSD) to decipher admixed GWAS from a population genetic perspective. The paper is largely divided into two sections. The first part explains why standard GWAS is statistically more powerful than Tractor by showing that the former benefits from allele frequency heterogeneity across ancestral source populations of the admixed population,

while the latter does not (Fig. 1A). We propose a simple improvement to Tractor that leverages existing meta-analysis techniques. Nevertheless, this improvement is still less powerful than standard GWAS as we show both theoretically and empirically. We also found that admixed cohorts offer higher power when using standard GWAS, compared to multi-ancestry cohort of the same size. It turns out that allele frequency heterogeneity increases power only in admixed cohorts and not in multi-ancestry cohorts comprised only of single-continental genomes.

The second half examines the predictions of the ePSD model which is central in modeling admixed genomes in admixed GWAS literature. To verify the model, we compared segments from single-continental genomes to their same-ancestry counterpart in admixed genomes (Fig. 1B). For example, we extracted African segments from African genomes and compared their linkage disequilibrium (LD) patterns to African segments from African-American (admixed) genomes. This is in contrast to a previous study that compared causal effect sizes across different ancestry segments by comparing segments of two different ancestries, namely, African and European segments (Additional file 1: Fig. S1) [26]. We found that the marginal effect sizes were different in same-ancestry segments that come from single-continental and admixed genomes, respectively. This was due to the distinct LD patterns in the two types of segments. We refer to The extended PSD and its connection to previous literature section for a further non-mathematical verbal discussion.

### The Pritchard-Stephens-Donnelly admixture model

The genotype of individual $i$ ($i = 1, \ldots, n_I$) is determined by a two-step process according to the Pritchard-Stephens-Donnelly (PSD) model. The global ancestry (or the ancestral proportions) $P_{il}$ is a vector of length $n_L$ (the number of ancestral source populations) that sums up to 1, i.e., $\sum_{l=1}^{n_L} P_{il} = 1$. The $l$th entry is the probability (hence, non-negative) that a randomly selected locus has originated from the $l$th ancestry. To meet this constraint, $P_{il}$ is commonly assumed to come from a Dirichlet distribution [1]. The full

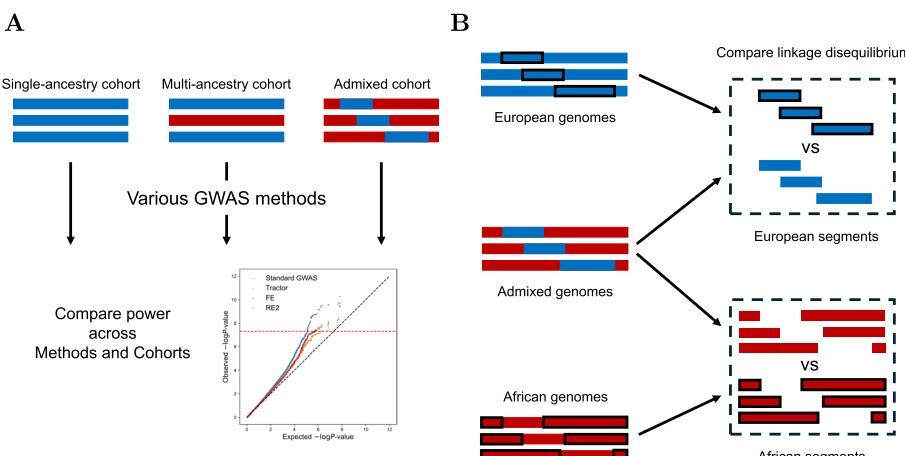

**Fig. 1** Overview of this study. **A** Power of various GWAS methods applied to different types of cohorts (single-ancestry, multi-ancestry, and admixed) were mathematically compared. **B** European segments from European genomes were compared to segments of the same ancestry from admixed genomes. The same comparison was made for African segments, too

distribution has little significance, and we are only interested in the first and the second moments of $P_{il}$. Briefly, all methods include global ancestry as a covariate, ruling out most of the global ancestry's variability. Only the first and the second moments of $\boldsymbol{P}_i$ will be mentioned in the paper without specifying the full distribution.

Local ancestry (LA) is assigned according to the probability specified by the global ancestry. At locus $k$, local ancestry $L_{ikl}$ counts how many copies of the locus originated from ancestry $l$. In diploids, including humans, $\sum_{l=1}^{n_L} L_{ikl} = 2$. It follows a multinomial distribution

$$L_{ik} \mid \boldsymbol{P}_i \sim \text{Multinomial}(\text{n} = 2, \text{p} = \boldsymbol{P}_i) \tag{1}$$

where $L_{ik} = [L_{ik1}, \ldots, L_{ikn_L}]^T$ and $P_i = [P_{i1}, \ldots, P_{in_L}]^T$. Precisely speaking, $\boldsymbol{L}_{ik}^h$ ($\boldsymbol{h} = \boldsymbol{m}$ for maternal and $\boldsymbol{h} = \boldsymbol{p}$ for paternal haplotypes) is sampled from $\boldsymbol{L}_{ik}^h \mid \boldsymbol{P}_i \sim \text{Multinomial}(\text{n} = 1, \text{p} = \boldsymbol{P}_i)$ and $\boldsymbol{L}_{ik} = \boldsymbol{L}_{ik}^m + \boldsymbol{L}_{ik}^p$. Finally, the genotype $G_{ik}^h = 0, 1$ of haplotype $\boldsymbol{h}$ is sampled from a Bernoulli distribution $G_{ik}^h \mid L_{ikl}^h = 1 \sim \text{Bernoulli}(p = f_{lk})$ conditional on $\boldsymbol{L}_{ik}^h$ where $l$ is the source ancestry of haplotype $h$ at locus $k$. $f_{lk}$ is the reference allele frequency at the locus in ancestry $l$.

### The extended PSD and its connection to previous literature

So far, we explained how genotype at a single locus is determined by the PSD model. The model does not describe the dependence between two or more loci and models the joint distribution as if the loci are mutually independent [1, 11]. Local ancestry inference methods, whether discriminative (RFMix) [8] or generative (hidden Markov model-based methods) [7, 10], employ an approximate coalescent with recombination model to describe the dependence between the loci [11, 27, 28]. In this framework, LD-related quantities of the ancestral and admixed populations such as LD covariance are realizations of the aforementioned coalescent process parameterized by the recombination rate and other evolutionary parameters [29], opposed to most, if not all, studies in GWAS literature that treat the LD-related quantities as a fixed parameter [30, 31].

To elaborate, modern admixed genomes are thought of as a realization of the evolutionary process that began at the time of admixture happened in the past. Coalescent with recombination explicitly models the recombination process backwards in time [32]. The observed LD patterns in the modern genome are merely one of the many possibilities that could have materialized from this random process [29]. Note that current allele frequencies of the source and admixed populations are also realizations of the random process in this setting due to genetic drift and mutation [7, 11]. In GWAS literature, we implicitly condition the current state of the population and treat the contemporary LD patterns as a parameter of the current population [30, 31]. For instance, the famous linkage disequilibrium score regression (LDSC) uses the sample LD scores estimated from the reference panel to approximate the population LD correlation of contemporary populations. It does not make any reference to the underlying evolutionary process of recombination. Only the realizations of the process as a collection of LD-related parameters are considered.

This divergence has led the complex trait literature on admixed populations to adopt an alternative model to describe the joint distribution of multiple loci [22–24]. We call this the extended PSD (ePSD) model. Following the convention of GWAS-related

studies, the model conditions on current populations and treats the current LD patterns as fixed parameters, rather than realizations of a random evolutionary process beginning in the past. It slightly relaxes the completely unlinked loci assumption of PSD by allowing the loci to be in LD with other variants in the nearby neighborhood. The neighborhood is stipulated to span substantially shorter than the length of local ancestry segments. This is roughly true in populations that experienced admixture only recently, as in the case of African Americans [24, 33, 34]. In this paper, we evaluate this assumption more thoroughly.

We make a precise description of the ePSD model. To address imperfect tagging, we use separate notations for causal variants $C_{ij}$ and markers $M_{ik}$. Indices $j$ and $k$ count the causal and marker variant loci, respectively. The ancestry-specific allele frequencies are $f_{lj} = \mathbb{P}(C_{ij}^{\boldsymbol{h}} = 1)$ and $g_{lk} = \mathbb{P}(M_{ik}^{\boldsymbol{h}} = 1)$ in ancestry $l$. We define the joint allele frequency of variants at loci $j$ and $k$ in ancestry $l$ as $h_{ljk} = P(C_{ij}^{\boldsymbol{h}} = 1, M_{ik}^{\boldsymbol{h}} = 1)$ in ancestry $l$. The linkage disequilibrium covariance of $l$th ancestry is then $D_{ljk} = h_{ljk} - f_{lj}g_{lk}$. For marker $k$, we can think of a set of causal variants linked to $k$, $[k]_l = \{j : D_{ljk} \neq 0\}$ specific to ancestry $l$. Let $[k] = \cup_{l=1}^{n_L}[k]_l$ be the union of all such variants. Assuming that the source populations have been well-mixing homogeneous populations under Hardy-Weinberg equilibrium (HWE) for a long time, compared to the time since admixture, $[k]_l$ for each ancestry $l$ only includes causal variants that are very close to $k$. Local ancestry segments are expected to be substantially longer because of the relatively few recombination that could have occurred since the recent admixture. Causal variants in $[k]_l$ are predominantly those that are in the same local ancestry segment as $k$. Hence, we can approximate the covariance between the variants $j$ and $k$ to be either one of $D_{ljk}$ for some ancestry $l$ when they are in the same ancestry segment or zero if they are in different segments, given that their local ancestry is known.

### What do standard GWAS and Tractor estimate under the PSD model?

We consider the following additive generative model for complex traits

$$Y_i = \sum_{j=1}^{n_J} C_{ij}\alpha_j + \alpha_0 + \varepsilon_i \tag{2}$$

where $Y_i$ is the trait, $C_{ij}$ are the causal variants, and $\varepsilon_i$ is the non-genetic error. $\alpha_0$ is the intercept and $\alpha_j$ are the causal effect sizes. Here, we assumed that the causal effects were the same across ancestry tracts [26]. Standard GWAS applies the following regression

$$Y_i \sim M_{ik}\beta_k + \beta_0 + \mathbf{X}_i\boldsymbol{\beta_X} \tag{3}$$

where $M_{ik}$ is the marker being tested and $\mathbf{X}_i$ is the set of covariates. $\beta_k$ is the marginal effect size. Covariates may include non-genetic variables like sex and age as well as global ancestry. As mentioned earlier, global ancestry and principal component (PC) covariates have a linear relationship between them [15, 16]. Therefore, when included in linear regression, they have the same effect.

Tractor separates marker reference allele count $M_{ik}$ according to the local ancestry to form $M_{ikl}$, the marker reference allele count that comes from the $l$th ancestry [20]. For example, suppose that an individual inherited exactly one copy from African and

European ancestry at locus $k$, i.e., $L_{ik1} = 1$ and $L_{ik2} = 1$. If the genotypes at both chromosomes at locus $k$ are the reference allele, $M_{ik1} = 1$ and $M_{ik2} = 1$. The total reference allele count is $M_{ik} = M_{ik1} + M_{ik2} = 2$. If only the African segment contained the reference allele, $M_{ik1} = 1$ and $M_{ik2} = 0$. Tractor's regression looks like

$$Y_i \sim \sum_{l=1}^{n_L} M_{ikl}\beta_{lk} + \sum_{l=2}^{n_L} L_{ikl}\gamma_{lk} + \beta_0 + \mathbf{X}_i\boldsymbol{\beta}_{\mathbf{X}} \tag{4}$$

It only includes $n_L - 1$ local ancestry variables because of the collinearity by $\sum_{l=1}^{n_L} L_{ikl} = 2$. Note that $M_{ikl}$ can never exceed $L_{ikl}$. $\beta_{lk}$ and $\gamma_{lk}$ are the ancestry-specific marginal effect size and the local ancestry effect size of the $l$th ancestry, respectively.

We can see that the linear equations of the generative model Eq. 2 and the regressions Eqs. 3 and 4 do not coincide. Hence, the connection between the marker coefficients $\beta_k$ and $\beta_{lk}$ to the causal effects $\alpha_j$ is obscure. A standard result is that $\beta_k$ is a linear combination of LD parameters and causal effect sizes [35–37]. For both ATT and Tractor, we can derive equations similar to the standard result. Under the generative model of Eq. 2, we can express the coefficients $\beta_k$ of ATT, $\beta_{lk}$, and $\gamma_{lk}$ of Tractor as a function of allele frequency, LD parameters, and $\alpha_j$. Here, we present the $n_L = 2$ case for exposition. Note that we are assuming the ePSD model here.

$$\beta_k = \sum_{j \in [k]} \underbrace{\frac{\mathbb{E}[\sum_l D_{ljk}P_{il}]}{\mathbb{E}[\sum_l g_{lk}(1-g_{lk})P_{il} + \sum_l g_{lk}^2 P_{il}(1-P_{il}) - \sum_{l \neq l'} g_{lk}g_{l'k}P_{il}P_{il'}]}}_{\text{Within-continental LD}} \alpha_j$$
$$+ \sum_{j=1}^{n_J} \underbrace{\frac{\sum_{l,l'} g_{lk}f_{l'j}\mathbb{E}[\text{Cov}(L_{ikl}^{\boldsymbol{h}}, L_{ijl'}^{\boldsymbol{h}} \mid \boldsymbol{P}_i)]}{\mathbb{E}[\sum_l g_{lk}(1-g_{lk})P_{il} + \sum_l g_{lk}^2 P_{il}(1-P_{il}) - \sum_{l \neq l'} g_{lk}g_{l'k}P_{il}P_{il'}]}}_{\text{Admixture LD}} \alpha_j \tag{5}$$

$$\beta_{lk} = \sum_{j \in [k]_l} \underbrace{\frac{D_{ljk}}{g_{lk}(1-g_{lk})}}_{\text{Within-continental LD}} \alpha_j \quad (l = 1, \dots, n_L) \tag{6}$$

$$\gamma_{lk} = \sum_{j \in [k]_l} \left( \frac{f_{lj} - h_{ljk}}{1 - g_{lk}} - \frac{f_{1j} - h_{1jk}}{1 - g_{1k}} \right) \alpha_j \quad (l = 2, \dots, n_L) \tag{7}$$

See Methods section for proofs. In Eqs. 5 and 6, both have contributions from within-continental LD. These are precisely the variants that the marker tags in the source populations. It is also interesting that the Tractor's coefficient $\beta_{lk}$ is the effect size one would expect from a single-continental GWAS applied to population $l$.

There are a few assumptions for our result to hold. One is that the covariate $\mathbf{X}_i$ accounts for the environmental confounding due to the correlation between marker $k$ and $\varepsilon_i$. Another assumption is that the population structure is solely due to admixture and that the ancestral source populations are in Hardy-Weinberg equilibrium (HWE). We also rule out important population phenomena such as assortative mating [38–40]. We focused on addressing genetic confounding in the context of admixture and its

interplay with various ancestry adjustments. Genetic confounding occurs when causal variants not tagged by the tested marker affect the trait [21, 25, 37, 41, 42]. This is unavoidable in the common univariate marginal testing procedures in GWAS. The tested variant can cover only a small portion of the genome, so the causal variants in the rest of the genome are left behind in the residuals. These distant causal variants can be correlated with the tested variant due to population structure, leading to spurious associations in the sense that the marker is picking up signals from remote regions that are far from its own position. This falls into the category of long-range LD due to population structure [11, 43].

In this context, Eqs. 6 and 7 show that the confounding due to remote causal variants is not accounted by local ancestry. Only the causal variants that are linked with marker $k$ are involved ($j \in [k]_l$). Rather, Eq. 7 suggests that local ancestry absorbs the remnant causal effect in which the marker did not fully tag. We can see this in the case of perfect linkage (or tagging) between $k$ and $j$. The coefficient is exactly zero because perfect tagging implies $f_{lj} = h_{ljk}$. Standard GWAS has an additional term due to admixture LD. $\mathrm{Cov}(L_{ikl}^{\hbar}, L_{ijl'}^{\hbar} \mid P_i)$ is the covariance between the local ancestry status between two loci $k$ and $j$. $L_{ikl}^{\hbar}$ and $L_{ijl'}^{\hbar}$ are independent draws given $P_i$ when the two loci are far apart, leading to zero covariance. For an adjacent pair of loci, especially if the admixture event was recent, this may not be true. Although it is impossible to express the covariance exclusively with LD parameters in source populations, this covariance decays slower than within-continental LD as a function of distance because fewer recombinations have taken place since admixture than in source populations. This explains why standard GWAS signals are less localized than Tractor signals in admixed GWAS. From this perspective, the claim that local ancestry corrects confounding should be understood that local ancestry removes signals from admixture LD, that persists longer than within-continental LD, and not the very remote signals from far away causal variants due to the global population structure [44].

### The Pritchard-Stephens-Donnelly model can predict GWAS power

Standard GWAS (the Armitage trend test, ATT) and Tractor are two popular methods for conducting GWAS in admixed populations. Simulation-based and empirical comparisons have previously been made in the literature provided the causal variant was directly tested [45, 46]. We complement the earlier findings with a precise mathematical formula. In the main text, we present the case of two source populations. See the Methods section for the general result of more than two source populations. When the tested variant is causal and does not tag any other variants, Eqs. 5 and 6 reduce to

$$\beta_k = \sum_{j=k} \alpha_j = \alpha_k \quad \text{and} \quad \beta_{lk} = \sum_{j=k} \alpha_j = \alpha_k \tag{8}$$

which are identical to the causal effect size of the tested variant.

A method is more powerful if the standard error is smaller provided the coefficients ($\beta_k$ and $\beta_{lk}$ in this case) are identical. We derived the standard errors of the methods using the PSD model. Let $M_{ik} = [M_{ik1}, \ldots M_{ikl}]^T$ and $\beta_k = [\beta_{1k}, \ldots, \beta_{n_Lk}]^T$. We also consider the centered genotypes $\tilde{M}_{ik} = M_{ik} - \mathbb{E}[M_{ik} \mid P_i]$, $\tilde{M}_{ikl} = M_{ikl} - \mathbb{E}[M_{ikl} \mid L_{ik}]$, and $\tilde{\mathbf{M}}_{ik} = M_{ik} - \mathbb{E}[M_{ik} \mid L_{ik}]$. Then, $\widehat{\beta}_k$ and $\widehat{\beta}_{lk}$'s standard errors are

$$\text{se}\left(\widehat{\beta}_k\right) \quad \propto \quad \sqrt{\frac{1}{2\sum_{l=1}^2 g_{lk}(1-g_{lk})\mathbb{E}[P_{il}] + 2(g_{1k}-g_{2k})^2\mathbb{E}[P_{i1}P_{i2}]}} \quad : \quad \text{Standard GWAS}$$

$$\text{se}\left(\widehat{\beta}_{lk}\right) \quad \propto \quad \sqrt{\frac{1}{2g_{lk}(1-g_{lk})\mathbb{E}[P_{il}]}} \quad : \quad \text{Tractor separate}$$

(9)

with the same proportional constants. We confirmed this formula by analyzing previously derived summary statistics of height and body mass index (BMI) in African-American individuals from the Population Architecture using Genomics and Epidemiology (PAGE) cohort [26, 47]. We extracted the standard errors from the summary statistics and compared them with the prediction of Eq. 9. We found that the predictions are extremely accurate (Fig. 2A). The result holds for other quantitative traits as well (Additional file 1: Figs. S2–S5).

Standard GWAS tests $\beta_k$, and Tractor can test each $\beta_{lk}$ separately or all $\boldsymbol{\beta}$ jointly. We can compare standard GWAS and separate ancestry-specific Tractor tests using Eq. 9 directly, but it requires some additional work to incorporate the combined Tractor test into the framework. Wald statistics, derived from Eq. 9, can incorporate the combined test. Note that Tractor is originally a likelihood-ratio test but is asymptotically equivalent to the Wald test [20]. The Wald test statistics divided by the sample size $n_I$ are approximately (in large $n_I$)

$$T_{n_I}^{\text{ATT}}/n_I \;\to\; \beta_k\mathbb{E}\left[\tilde{M}_{ik}^2\right]\beta_k = \beta_k^2\mathbb{E}\left[\tilde{M}_{ik}^2\right]\cdot C \quad : \quad \text{Standard GWAS}$$

$$T_{n_I}^{l,\text{Tractor}}/n_I \;\to\; \beta_{lk}\mathbb{E}\left[\tilde{M}_{ikl}^2\right]\beta_{lk} = \beta_k^2\mathbb{E}\left[\tilde{M}_{ikl}^2\right]\cdot C \quad : \quad \text{Tractor separate}$$

$$T_{n_I}^{\text{Tractor}}/n_I \;\to\; \boldsymbol{\beta}_k^T\mathbb{E}\left[\tilde{\mathbf{M}}_{ik}^T\tilde{\mathbf{M}}_{ik}\right]\boldsymbol{\beta}_k = \beta_k^2\mathbf{1}_{n_L}^T\mathbb{E}\left[\tilde{\mathbf{M}}_{ik}^T\tilde{\mathbf{M}}_{ik}\right]\mathbf{1}_{n_L}\cdot C \quad : \quad \text{Tractor combined}$$

(10)

for some common constant $C > 0$ and $\mathbf{1}_{n_L}$ is a vector of ones of length $n_L$ (see Methods section). Recall that we assumed equal effect sizes across ancestries. The expectations $\mathbb{E}\left[\tilde{M}_{ik}^2\right]$, $\mathbb{E}\left[\tilde{M}_{ikl}^2\right]$, and $\mathbb{E}\left[\tilde{\mathbf{M}}_{ik}^T\tilde{\mathbf{M}}_{ik}\right]$ can be deduced from the PSD model, which is our core contribution.

$$\mathbb{E}\left[\tilde{M}_{ik}^2\right] = 2\sum_{l=1}^2 g_{lk}(1-g_{lk})\mathbb{E}[P_{il}] + 2(g_{1k}-g_{2k})^2\mathbb{E}[P_{i1}P_{i2}]$$

$$\mathbb{E}\left[\tilde{\mathbf{M}}_{ik}^T\tilde{\mathbf{M}}_{ik}\right] = \begin{bmatrix} 2g_{1k}(1-g_{1k})\mathbb{E}[P_{i1}] & 0 \\ 0 & 2g_{2k}(1-g_{2k})\mathbb{E}[P_{i2}] \end{bmatrix}$$

(11)

We highlight that the last matrix is diagonal, which is an important observation that will be repeatedly used. This is surprising because it implies that the two regression estimates $\widehat{\beta}_{1k}$ and $\widehat{\beta}_{2k}$ from the same regression Eq. 4 on the same data are independent. We can then see that the statistics are ordered (in increasing order)

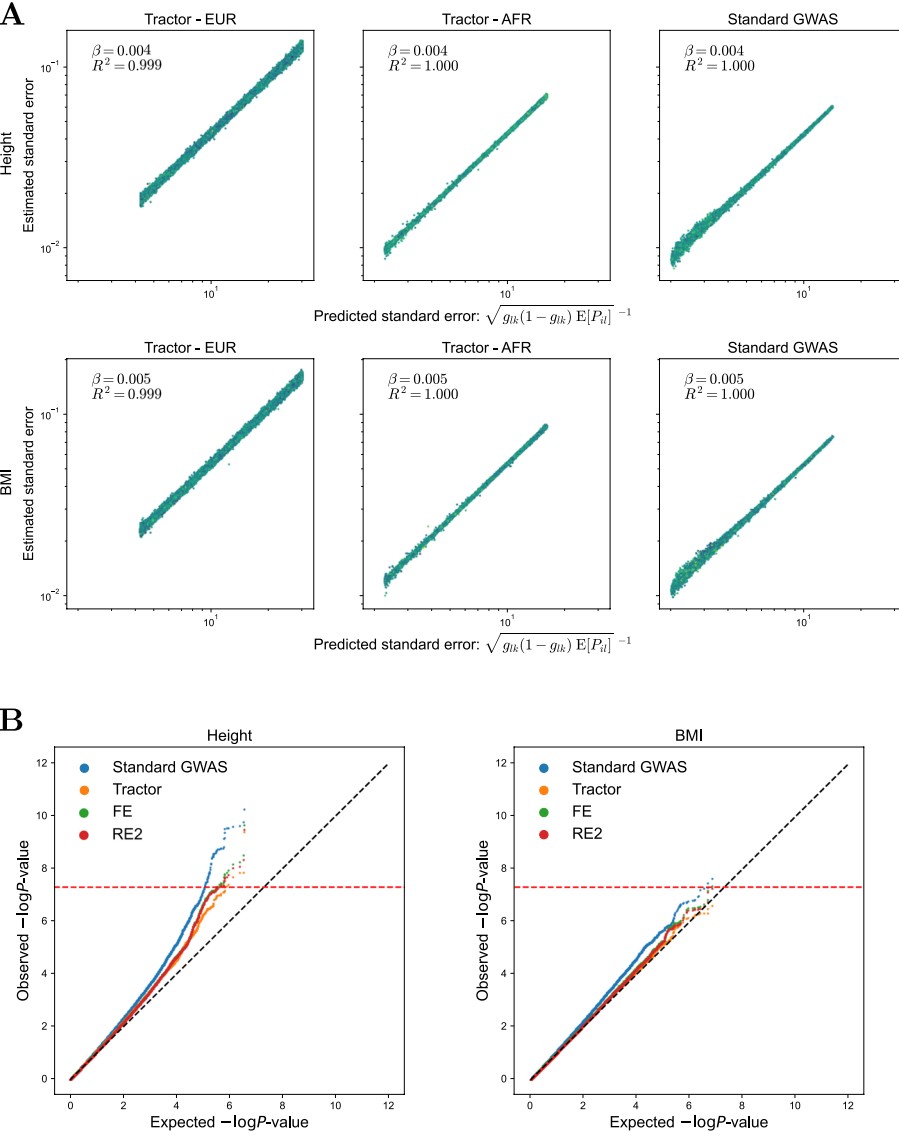

**Fig. 2** Predictions of the PSD model evaluated in real data. **A** Comparison of predicted and estimated standard error of regression coefficients. The top panel is for height and the lower panel is for body mass index (BMI). **B** Quantile-quantile (QQ) plots of GWAS results of height (left) and BMI (right) in the PAGE cohort. FE: fixed-effects meta-analysis, RE2: Han-Eskin random-effects meta-analysis

$$
\begin{aligned}
T_{n_I}^{l,\text{Tractor}}/n_I &= \beta_k^2 \cdot 2g_{lk}(1-g_{lk})\mathbb{E}[P_{il}] \cdot C \\
&\leq \quad T_{n_I}^{\text{Tractor}}/n_I = \beta_k^2 \cdot 2\sum_{l=1}^{2} g_{lk}(1-g_{lk})\mathbb{E}[P_{il}] \cdot C \\
&\leq \quad T_{n_I}^{\text{ATT}}/n_I = \beta_k^2 \cdot \left[ 2\sum_{l=1}^{2} g_{lk}(1-g_{lk})\mathbb{E}[P_{il}] + 2(g_{1k}-g_{2k})^2\mathbb{E}[P_{i1}P_{i2}] \right] \cdot C
\end{aligned}
\tag{12}
$$

ATT's (the standard GWAS) test statistics is always larger or equal to the Tractor's statistics. This advantage is driven by the allele frequency difference $g_{1k} - g_{2k}$ between the two source populations. This part explains how admixture LD contributes to power. The coefficient being tested remains the same, but admixture LD increases the test

statistics by improving the precision of the estimate, i.e., a smaller standard error and a narrower confidence interval. The combined statistics $T_{n_I}^{\text{Tractor}}$ additionally suffers from the increased degrees-of-freedom of the null $\chi^2$-distribution. What is new is that Tractor is less powerful than standard GWAS even without the effect of increased degrees-of-freedom because the absolute value of the test statistics is smaller. Also, Tractor does not benefit from allele frequency heterogeneity across ancestral populations at all, although it still does from heterogeneous effect sizes. This means that admixed GWAS is unlikely to add more statistical power than single-continental GWAS if one uses Tractor in the absence of a conspicuous causal effect size heterogeneity. The gain of ATT from $(g_{1k} - g_{2k})^2 \mathbb{E}[P_{i1}P_{i2}]$ depends on $\mathbb{E}[P_{i1}P_{i2}] = \mathbb{E}[P_{i1}(1 - P_{i1})]$, which is larger when individuals with equal ancestral proportions from both ancestries are common in the population (function $f(x) = x(1 - x)$ is maximized at $x = 0.5$). It is important to note that the power gain driven by $(g_{1k} - g_{2k})^2 \mathbb{E}[P_{i1}P_{i2}]$ is absent in a multi-ancestry cohort only made up of people of single-continental origins. People from ancestry $l$ will have $P_{il} = 1$ and $P_{il'} = 0$ for $l' \neq l$, so $\mathbb{E}[P_{il}P_{i2}]$ is always zero.

A more direct evaluation of the claims on statistical power can be made by the quantile-quantile (QQ) plot of $P$ values (Fig. 2B). ATT was more powerful than Tractor's combined test, as shown previously [45, 46]. We have mentioned that this follows from two factors, namely, the smaller test statistics and the larger degree-of-freedom of Tractor than ATT. Although the first factor cannot be resolved, we can overcome the second issue while keeping the test statistics constant.

The ancestry-specific test statistics of Tractor $T_{n_I}^{l,\text{Tractor}}$ are mutually independent for all pairs of $l = 1, \ldots, n_L$ (Eq. 11 and Methods section). This is surprising because all the ancestry-specific statistics are obtained from the same regression and data. The independence allows us to combine them using existing meta-analysis methods that combine independent summary statistics. The widely adopted fixed-effects (FE) meta-analysis produces a test statistics equal to the Tractor's combined test [48, 49]. Despite having the same test statistics value, it only has one degrees-of-freedom because it tests a different hypothesis. FE meta-analysis tests a single-parameter hypothesis ($\mathbb{H}_0 : \beta_k = 0$) assuming that all ancestry-specific effect sizes are equal, and Tractor tests a $n_L$-parameter hypothesis that allows all effect sizes to vary ($\mathbb{H}_0 : \beta_{lk} = 0$ for all $l = 1, \ldots, n_L$). We applied another meta-analysis based on the Han-Eskin model, commonly abbreviated as RE2 (random-effects two) [50, 51]. When applied to the same PAGE summary statistics of height and BMI, we found that FE and RE2 achieve better power than Tractor's combined test (Fig. 2B). RE2 has a 1.5 degrees-of-freedom and is known to be powerful when the effect size estimates are heterogeneous. The fact that it performs worse than ATT and similarly to FE suggests that the marginal effect size heterogeneity is not large in the two traits. The results remain the same in other quantitative traits as well (Additional file 1: Figs. S6 and S7).

### Testing the extended PSD model in real data and simulations

As Eq. 6 is deduced from the ePSD model, we can indirectly assess the model by testing Eq. 6. The equation says that Tractor's ancestry-specific estimates are identical to the summary statistics had the GWAS conducted on single-continental genomes. Given that the prediction is correct, we hypothesized that measuring the genetic correlation of

Tractor's African and European effect sizes with the corresponding single-continental summary statistics will produce genetic correlations close to 1 provided the ePSD is correct. Tractor estimates (summary statistics) were obtained from the PAGE cohort and the single-continental summary statistics were obtained from PanUKBB [26, 47, 52]. We used linkage disequilibrium score regression (LDSC) to estimate the genetic correlation (see Methods section).

In 15 quantitative traits, the frequent appearance of negative heritability estimates produced invalid genetic correlations (Fig. 3A). Such traits include hemoglobin 1Ac (Hb1Ac), C-reactive proteins (CRP), diastolic blood pressure (DBP), estimated glomerular filtration rate (eGFR), fasting glucose, height, platelet count, and waist-to-hip ratio (WHR). In some traits, such as CRP, high-density cholesterol (HDL), low-density cholesterol (LDL), systolic blood pressure (SBP), and WHR, genetic correlations of European marginal effect sizes were significantly lower than 1. However, the confidence intervals were often too wide, especially in African marginal effect sizes, to draw reliable conclusions.

LDSC measures the genetic correlation by recognizing that marginal effect sizes are a weighted sum of causal effects weighted by LD correlation (as in Eqs. 5 and 6) of causal effects. Since causal effect size is rarely observed directly, LDSC disentangles the causal signals from the marginal effect sizes by using the LD scores that determine the relationship between causal and marginal effects [30]. Therefore, the low genetic correlation could be explained by either different causal effect sizes or LD patterns. We investigated the latter factor through simulations as the former reason has been studied elsewhere [26]. Also, it was ePSD's prediction that the local LD patterns of an admixed genome should be the same as those of the ancestral source populations. We return to the precise definition of *local* shortly which is roughly around the scale of 0.1 cM. We conducted simulations to dig into the issue.

We simulated a total of 20,000 individuals using `msprime`, a coalescent-based simulator [53] (see Methods section). 5000, 5000, and 10,000 African, European, and African American were simulated, respectively, using the American admixture demographic model described previously after dropping East Asian ancestry [54–56] (Additional file 1: Fig. S8). The correlation between Tractor estimates and single-continental marginal effect sizes was very weak (Fig. 3B). This result was immediately against the prediction of Eq. 6. We observed the ePSD model to hold locally where marginal effect sizes at the causal loci were highly concordant (Fig. 3C). Nevertheless, the concordance dropped with the increasing proportion of causal loci, or higher polygenicity. This implied that the local LD between the tested causal variant and other nearby causal variants in admixed populations was different from the single-continental counterpart.

Indeed, comparing normalized LD covariances (Eqs. 13 and 14) of admixed and single-continental genomes revealed only moderately concordant patterns. Here, we compared African (European) segments from admixed individuals to the corresponding segments in single-continental (non-admixed) African (European) genomes, which is different from a previous study that compared segments of different ancestries (African versus Europeans) all obtained from admixed individuals [26]. We binned the pairs of loci according to their physical distance (Fig. 4A). The correlation was fairly high within the range (~0.1 cM or 100 kb) of within-continental LD blocks [57, 58], but considerably

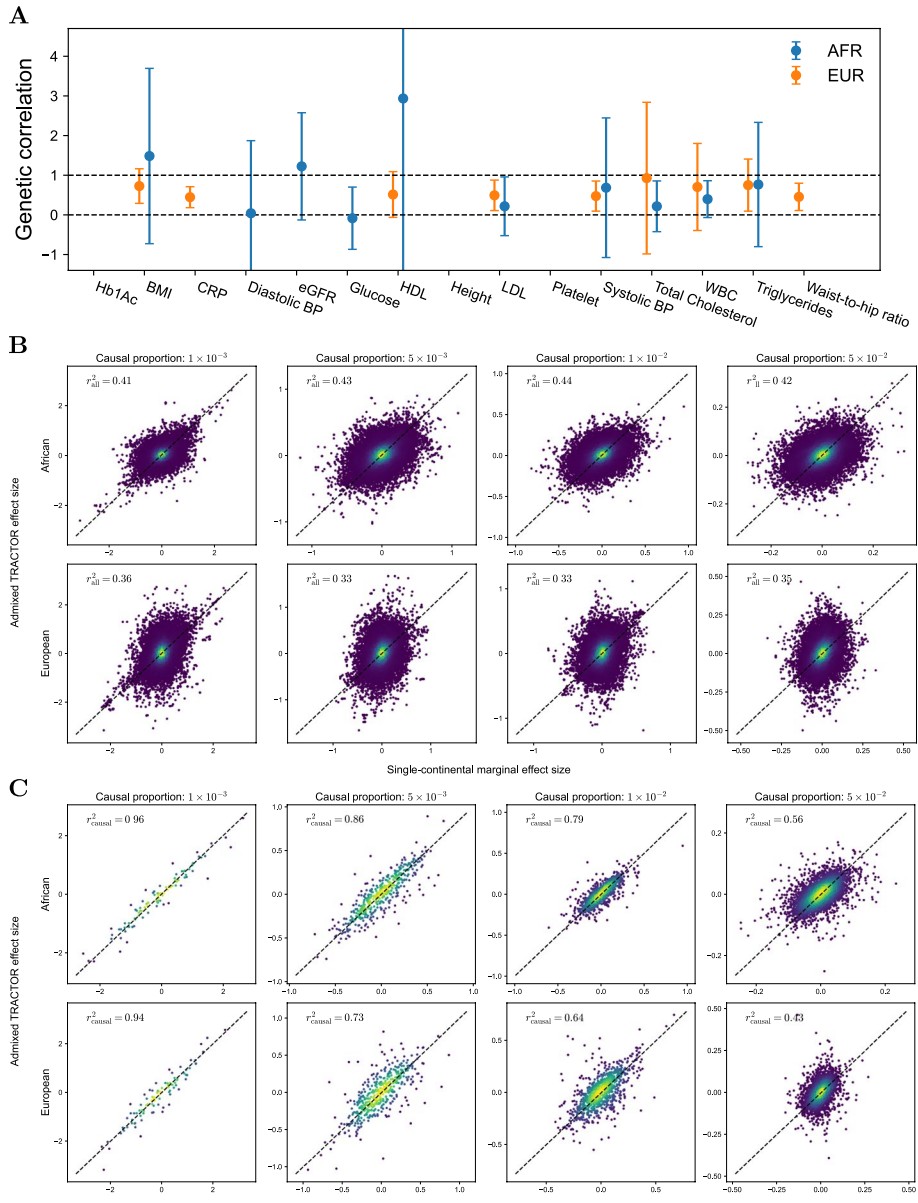

**Fig. 3** Predictions of the ePSD model evaluated in real and simulated data. **A** The genetic correlation of 15 traits and their 95% confidence intervals estimated by LDSC. Invalid estimates were omitted from the plot. The error bar of HDL was truncated due to its wide margin. **B** Marginal effect sizes of Tractor and single-continental GWAS in simulated African and European genomes. Brighter means a higher density of points. **C** Same as **B** but only causal variants were plotted. Note that African segments from admixed and single-continental genomes were compared, and the same for European segments

low in the ranges (∼1 cM or 1000 kb) considered in LD scores [30]. The correlation's decay were faster in European ancestry that had lower occupancy in the genome (African:European = 8:2). This is likely because the LA segments from the minority population is more easily surrounded by the majority LA segments.

The LD covariances between two loci $k$ and $j$ of the African ($l = 1$) and European ancestry ($l = 2$) in admixed genome are the coefficients of $M_{ik1}$ and $M_{ik2}$ in regression

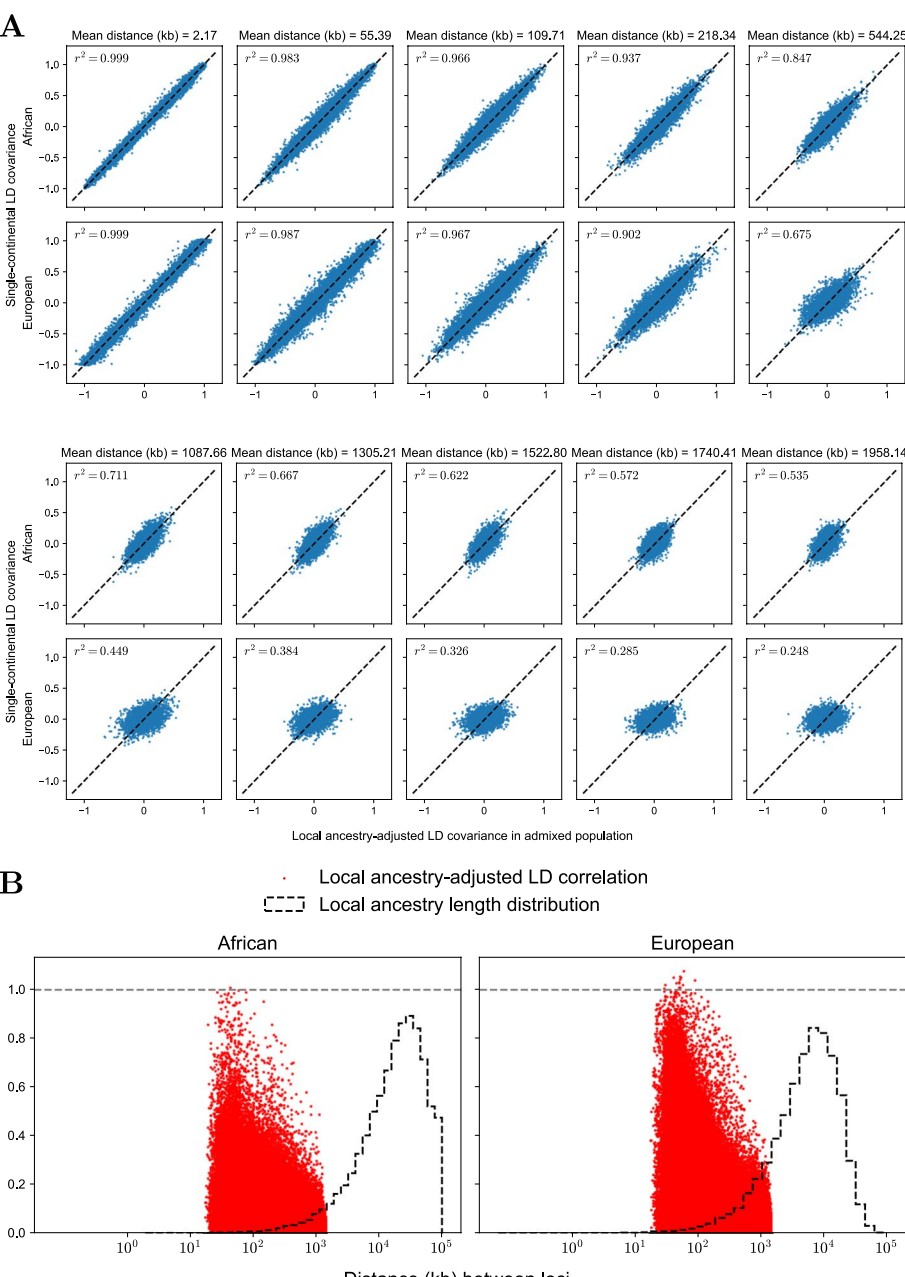

**Fig. 4** Direct evaluation of the ePSD model in simulated data. **A** *y*-axis is the LD covariance obtained from single continental genomes and *x*-axis is the LD covariance obtained from admixed genomes. The first row compares the LD patterns in African segments and the second row compares those from European segments. **B** Scatterplots of local-ancestry adjusted LD correlation versus physical distance were laid over the histogram of local ancestry segment lengths. The left and the right panel are from African and European segments, respectively

$$C_{ij} \sim \{\text{African covariance}\} \cdot M_{ik1} + \{\text{European covariance}\} \cdot M_{ik2} + \{\text{some coefficient}\} \cdot L_{ik2}$$
(13)

In single-continental genomes, the covariances are simply the coefficient of $M_{ik}$ in regression

$$C_{ij} \sim \{\text{African or European covariance}\} \cdot M_{ik}$$
(14)

for each ancestry. The covariances of the same ancestry in Eqs. 13 and 14 should be the same under the ePSD model. Namely, they should be equal to

$$\frac{D_{ljk}}{g_{lk}(1 - g_{lk})}$$
(15)

for each ancestry $l$, which is the weights of the causal effect sizes $\alpha_j$ in Eq. 6.

This finding explains the large disagreement between the Tractor and single-continental marginal estimates. Since the marginal effect sizes are a sum of these LD covariances multiplied by the causal effect sizes (Eq. 6), adding many modestly discordant values will lead to highly discordant quantities overall. This is pronounced when the polygenicity is high so the number of loci entering the sum is larger (Fig. 3C). We also computed the LD correlation from the covariances in Eq. 13 by multiplying $\sqrt{\frac{g_{lk}(1-g_{lk})}{f_{lj}(1-f_{lj})}}$, which should yield

$$\frac{D_{ljk}}{\sqrt{g_{lk}(1 - g_{lk})f_{lj}(1 - f_{lj})}}$$
(16)

provided the ePSD is correct. Comparing the ancestry segment lengths and LD coefficient shows that the LD's extent is not necessarily shorter than local ancestry segments (Fig. 4B). The overlap between the segment lengths and LD coefficient is larger in Europeans, which explains the low concordance in Europeans in Fig. 4A.

In sum, what we saw in LDSC analysis of Tractor and within-continental summary statistics is explained by the differences in local LD patterns between chromosomal segments of the same ancestral origin from admixed and single-continental genomes. Although the pattern is highly concordant in very short regions, it differentiates quickly as the region grows in length. Hence, ePSD turns out to be a good approximation in short regions but performs poorly on longer scales that are as short as 1000 kb (or 1 cM).

## Discussion

In this work, we mathematically derived identifiable predictions of the PSD model and its extension. These predictions illuminate the properties of a variety of GWAS methods applied to admixed genomes and suggest simple but effective improvements. The theory explains why standard GWAS regression is more powerful than Tractor by showing that only standard GWAS benefits from allele frequency heterogeneity across source populations. In contrast to the original PSD model, however, we found that the extended PSD model only remains valid in very short regions in the genome.

There have been several studies comparing the power of various GWAS methods applied to admixed cohorts [45, 59]. The standard GWAS regression, often called the Armitage trend test (ATT), has been found to be the most powerful across various settings. Our theoretical analysis reconfirms this claim by showing that GWAS estimates'

standard error is enlarged when adjusting local ancestry as in Tractor. We then showed that the relative power loss of Tractor can be partially ameliorated by combining ancestry-specific estimates through meta-analysis. This was possible because Tractor's estimates were independent despite being obtained from the same data given the PSD model. These findings were highly concordant with real data.

Our study raises caution to the common assumption that LA segments extend beyond the range of within-continental LD. The assumption greatly simplifies the LD structure of admixed genome by conferring independence between variants on different LA segments conditional on LA. In GWAS, this allows the isolation of ancestry-specific marginal effect size by confining the LD within the LA segment [24]. In polygenic risk score prediction, variants can be assigned ancestry-specific weights based solely on their segment ancestry [22, 23]. In other words, summary statistics from single-continental GWAS can be used to construct risk scores for admixed individuals. This modeling strategy is further supported by our mathematical result deduced from ePSD: ancestry-specific marginal effect sizes in admixed genomes are identical to the marginal effect sizes obtained from the corresponding single-continental genomes.

Nevertheless, the assumption fails to explain the findings in real and simulated data. Firstly, heritability estimates (required for genetic correlation) were frequently negative, indicating a mismatch between single-continental reference LD and admixed LD. Secondly, in simulations, we were able to observe the overlap between the local ancestry length distribution and LD. This translated into a low correlation between marginal effect sizes from single-continental and admixed genomes. By comparing segments of the same ancestry extracted from admixed and single-continental genomes, we showed that their LD pattern can differ substantially. We emphasize that this comparison is different from a previous study comparing segments of different ancestry (e.g., African versus European) solely from admixed genomes to quantify the causal effect size heterogeneity [26]. Instead, we compared segments coming from the same ancestry and saw if the LD patterns differed depending on whether they came from single-continental genomes or admixed genomes. This means that marginal effect sizes of the same ancestry can still be different without causal effect heterogeneity solely due to LD heterogeneity.

Then why did the assumption seem to be successful in previous studies? It is likely that the overly simplistic simulation design has been the problem. For example, the ePSD model is part of the simulation in the original Tractor paper [20]. Relatively more accurate simulation algorithms based on the classic coalescent still fail because they cannot reproduce long-range LD that extends beyond LA segments [60]. Indeed, we find in simulations that even a simple model of a single admixture event can produce long-range LD patterns that last more than 10 generations. Under more realistic models where migrations continues to the present, LD is likely to last longer [60, 61].

It is worth noting that our results do not entirely discourage the use of the methods that implicitly assume the ePSD model. For example, for risk prediction purposes, the practical utility and performance of a method are not disqualified by its imperfect modeling assumption [22, 23]. We only raise caution on drawing scientific conclusions at a genome-wide scale based on the ePSD model as the model remains valid in very short regions shorter than 500 kb (or 0.5 cM).

There are several caveats in our analysis. Firstly, the wide confidence intervals of genetic correlation analysis leave a large uncertainty in the analysis. The wide intervals stem from the small effective population size of the European portion of admixed genomes and the African participants of the PanUKBB. Therefore, a larger cohort of admixed and African participants is required to address the issue fully. Nevertheless, the simulation shows that the low genetic correlation found in real data is likely to remain in larger data. Secondly, the low genetic correlation between admixed African GWAS and standard African GWAS may come from their true difference. African genomes exhibit a substantially higher diversity level than other continental counterparts [62–64]. Hence, the true genetic difference between the genomes may have caused the low genetic correlation. Finally, the current analysis is confined to quantitative traits and does not consider binary phenotypes based on logit and probit models. Future work should investigate this issue by extending the framework proposed in this paper.

## Conclusions

The Pritchard-Stephens-Donnelly (PSD) model produces highly accurate predictions for the power of various GWAS methods conducted on admixed genomes. Notably, an admixed cohort is likely to offer higher power than a multi-ancestry cohort of the same size consisting of single-continental individuals. This benefit applies only to standard GWAS, not to Tractor. The model's popular extension, which we call the extended PSD (ePSD) model, hinges on the assumption that local linkage disequilibrium is confined within local ancestry segments. Nevertheless, we show with both simulations and real data that this assumption does not hold in practice. As a result, admixed and single-continental genome segments exhibit distinct LD patterns. This finding raises caution about the widespread use of the ePSD model and calls for a better model of genetic recombination.

## Methods

The relevant scripts and codes can be found at the Zenodo repository (https://zenodo.org/records/15637591) [65].

### PAGE and UK Biobank summary statistics

We analyzed summary statistics of one dataset of African-European admixed individuals. Population Architecture through Genomics and Environment (PAGE) study included 17,299 genotyped individuals with African-European admixed ancestries determined by estimated admixture proportion and with approximately 6.9 million variants. Detailed steps of quality control and processing, including GWAS using `admix-kit` and local ancestry inference using `RFMix`, can be found in Genotype data processing section of Hou and colleagues [26]. Summary statistics of 15 traits of UK Biobank participants of African and European ancestry were downloaded from the Pan UK Biobank repository (https://pan.ukbb.broadinstitute.org/) [52].

### Marginal effect size of standard GWAS and Tractor

Most of the mathematical result is obtained by computing the (conditional) expectation and (co)variance of $M_{ik}$, $M_{ikl}$, and $C_{ij}$. The calculations are shared across proofs and can

be found in Additional file 2: Supplementary Note. In the calculations, we explain how the assumptions

- No residual environmental confounding: $M_{ik}, C_{ij} \perp\!\!\!\perp \varepsilon_i \mid \boldsymbol{P}_i$
- Hardy-Weinberg equilibrium in source populations: $M_{ik}^{\boldsymbol{m}} \perp\!\!\!\perp M_{ik}^{\boldsymbol{p}} \mid \boldsymbol{L}_{ik}$

lead to the formulas derived in this paper. In the following, we briefly explain how the results were derived from the results in Additional file 2: Supplementary Note.

Equation 5 follows from the formula

$$\beta_k = \sum_{j=1}^{n_J} \frac{\mathbb{E}[\text{Cov}(C_{ij}, M_{ik} \mid \boldsymbol{P}_i)]}{\mathbb{E}[\text{Var}(M_{ik} \mid \boldsymbol{P}_i)]} \alpha_j \tag{17}$$

that follows from linear regression theory [66, 67]. As we show in Additional file 2: Supplementary Note,

$$\text{Var}(M_{ik} \mid \boldsymbol{P}_i) = 2\sum_{l=1}^{n_L} g_{lk}(1-g_{lk})P_{il} + 2\sum_{l=1}^{n_L} g_{lk}^2 P_{il}(1-P_{il}) - 2\sum_{l\neq l'} g_{lk}g_{l'k}P_{il}P_{il'}$$

$$\text{Cov}(C_{ij}, M_{ik} \mid \boldsymbol{P}_i) = 2\sum_{l=1}^{n_L} D_{ljk}P_{il} + 2\sum_{l=1}^{n_L} g_{lk}f_{lj}P_{il}(1-P_{il}) - 2\sum_{l\neq l'} g_{lk}f_{l'j}P_{il}P_{il'} \tag{18}$$

and substituting these equations to Eq. 17 proves Eq. 5.

Equations 6 and 7 follow from the following statement

$$\begin{aligned}
\mathbb{E}[Y_i \mid \boldsymbol{M}_{ik}, \boldsymbol{L}_{ik}, \boldsymbol{P}_i] = &\sum_{l=1}^{n_L} M_{ikl} \sum_{j\in[k]_l} \frac{D_{ljk}}{g_{lk}(1-g_{lk})} \alpha_j \\
&+ \sum_{l=2}^{n_L} L_{ikl} \sum_{j\in[k]_l} \left( \frac{f_{lj}-h_{ljk}}{1-g_k} - \frac{f_{1j}-h_{1jk}}{1-g_{1k}} \right) \alpha_j \\
&+ \sum_{l=2}^{n_L} P_{il}\delta_{lk} + \beta_0
\end{aligned} \tag{19}$$

where $\delta_{lk}$ $(l = 2, \ldots, n_L)$ and $\beta_0$ are some constants. When the conditional expectation of the response variable $Y_i$ is linear in respect to the conditional variables $\boldsymbol{M}_{ik}, \boldsymbol{L}_{ik}$, and $\boldsymbol{P}_i$, linear regression's estimated coefficients are unbiased for the coefficients of the conditional variables [66].

### Wald test statistics and GWAS standard errors

Buja et al. [68] and Ding [69] proved the asymptotic distribution of linear regression coefficients given the covariates. If we assume that the amount of trait variance explained by a single locus is small, their formula becomes

$$\sqrt{n_I}\left(\widehat{\beta}_k - \beta_k\right) \xrightarrow[n_I \to \infty]{} \mathcal{N}\left(\mathbf{0}, \mathbb{E}\left[\tilde{M}_{ik}^2\right]^{-1} \cdot \mathbb{E}[\text{Var}(Y_i \mid \boldsymbol{P}_i)]\right)$$
$$\sqrt{n_I}\left(\widehat{\boldsymbol{\beta}}_k - \boldsymbol{\beta}_k\right) \xrightarrow[n_I \to \infty]{} \mathcal{N}\left(\mathbf{0}, \mathbb{E}\left[\tilde{\mathbf{M}}_{ik}^T \tilde{\mathbf{M}}_{ik}\right]^{-1} \cdot \mathbb{E}[\text{Var}(Y_i \mid \boldsymbol{P}_i)]\right) \quad (20)$$

The standard errors of $\widehat{\boldsymbol{\beta}}_k = [\widehat{\beta}_{1k}, \dots, \widehat{\beta}_{n_L k}]^T$ and $\widehat{\beta}_k$ are, therefore, the square root of the right-hand side. This, together with Eq. 11, gives Eq. 9 where the omitted proportional constant is $\mathbb{E}[\text{Var}(Y_i \mid \boldsymbol{P}_i)]$. Proof of Eq. 11 is in Additional file 2: Supplementary Note.

Wald statistics is simply a square of regression estimators in Eq. 20 normalized by the covariance matrix on the right-hand side.

$$T_{n_I}^{\text{ATT}} = n_I \left(\widehat{\beta}_k - \beta_k\right)^T \mathbb{E}\left[\tilde{M}_{ik}^2\right] \left(\widehat{\beta}_k - \beta_k\right) \cdot \mathbb{E}[\text{Var}(Y_i \mid \boldsymbol{P}_i)]^{-1}$$
$$T_{n_I}^{\text{Tractor}} = n_I \left(\widehat{\boldsymbol{\beta}}_k - \boldsymbol{\beta}_k\right)^T \mathbb{E}\left[\tilde{\mathbf{M}}_{ik}^T \tilde{\mathbf{M}}_{ik}\right] \left(\widehat{\boldsymbol{\beta}}_k - \boldsymbol{\beta}_k\right) \cdot \mathbb{E}[\text{Var}(Y_i \mid \boldsymbol{P}_i)]^{-1} \quad (21)$$

The statistics calculated assuming the null ($\boldsymbol{\beta}_k = \mathbf{0}$ and $\beta_k = 0$) are

$$T_{n_I}^{\text{ATT}}/n_I = \widehat{\beta}_k^T \mathbb{E}\left[\tilde{M}_{ik}^2\right] \widehat{\beta}_k \cdot \mathbb{E}[\text{Var}(Y_i \mid \boldsymbol{P}_i)]^{-1} \rightarrow \beta_k^T \mathbb{E}\left[\tilde{M}_{ik}^2\right] \beta_k \cdot \mathbb{E}[\text{Var}(Y_i \mid \boldsymbol{P}_i)]^{-1}$$
$$T_{n_I}^{\text{Tractor}}/n_I = \widehat{\boldsymbol{\beta}}_k^T \mathbb{E}\left[\tilde{\mathbf{M}}_{ik}^T \tilde{\mathbf{M}}_{ik}\right] \widehat{\boldsymbol{\beta}}_k \cdot \mathbb{E}[\text{Var}(Y_i \mid \boldsymbol{P}_i)]^{-1} \rightarrow \boldsymbol{\beta}_k^T \mathbb{E}\left[\tilde{\mathbf{M}}_{ik}^T \tilde{\mathbf{M}}_{ik}\right] \boldsymbol{\beta}_k \cdot \mathbb{E}[\text{Var}(Y_i \mid \boldsymbol{P}_i)]^{-1}$$
$$(22)$$

which converges to Eq. 10 under the alternative hypothesis in large $n_I$.

### Tractor summary statistics meta-analysis

Fixed-effects and random-effects (RE2) meta-analysis were performed using the RE2C software (https://github.com/cuelee/RE2C) [50, 51]. Ancestry-specific summary statistics were supplied to RE2C software to produce FE and RE2 meta-analysis *P* values.

### Linkage disequilibrium score regression using Tractor estimates

The standard error of single-continental GWAS is

$$\sqrt{\frac{1}{2g_{lk}(1 - g_{lk})}} \quad (23)$$

Equation 9 implies that the standard error of the *l*th ancestry-specific Tractor estimate is enlarged ($\because P_{il} \le 1$) by a factor of

$$\sqrt{\frac{1}{\mathbb{E}[P_{il}]}} \quad (24)$$

which means that the *Z*-score is reduced by a factor of

$$\sqrt{\mathbb{E}[P_{il}]} \quad (25)$$

compared to a single-continental GWAS of the same ancestry and sample size. Therefore, the effective sample size is the admixed sample size times Eq. 25. African and European LD score was obtained from Pan UK Biobank (https://pan.ukbb.broadinstitute.org/downloads/index.html) [52].

### African-American genome simulation

The demographic model of African Americans was retrieved from `stdpopsim` 0.2.0 catalog [56]. The model id we used was `AmericanAdmixture_4B11` that was reported in Browning and colleagues [54]. We excluded the East Asian contribution and modified the admixture proportion of admixed African Americans to be 80% African and 20% Europeans. We assumed that the admixture occurred 12 generations ago [24, 33, 34]. The resulting demography model file is available in our paper's github in `demes` format [70]. A visual presentation of the demographic model is in Additional file 1: Fig. S8.

Using the above demography, we simulated 5000, 5000, and 10,000 African, European, and African American individuals with `msprime` 1.3.3 [53, 71]. The genome length was set to $10^8$ base pairs (approximately half the length of chromosome 3). Recombination and mutation rate were set to $1.15 \times 10^{-8}$ and $1.29 \times 10^{-8}$ which were adapted from *Homo sapiens* stdpopsim catalog [56]. Following the suggestion from Nelson and colleagues, the first 5 generations (backwards in time) were set to follow a Wright-Fisher process [60]. For the remaining period in the past, the default coalescent process of `msprime` was used. Local ancestry information of African Americans were extracted from the tree sequence generated by `msprime` using `tspop`. By recording all the lineages in the tree sequence that were present right before the admixture event, `tspop` tracks the ancestry segments of the modern genome to their ancestors in the ancestral source populations [72].

### Trait simulation and LD covariance computation

We simulated traits from the simulated genomes using `tstrait` [73]. It randomly selects sites from the tree sequence storing the genome simulated from `msprime` with a probability given by the user (i.e., polygenicity or causal proportion) and assigns an effect size drawn from a normal distribution. The normal distribution's variance is set by the heritability parameter that ranges from 0 to 1. Because our goal was to obtain true effect sizes from the simulated data, we simulated the trait with heritability 1, which means that no environmental noise was injected to the simulation.

The polygenicity parameters used were $1 \times 10^{-3}$, $5 \times 10^{-3}$, $1 \times 10^{-2}$, and $5 \times 10^{-2}$. After simulating the trait given a polygenicity, we regressed the trait by Eq. 4 to obtain the coefficients in admixed genomes. In single-continental genomes, we simply regressed the trait on $M_{ik}$ to get the coefficients with Eq. 3. We compared the coefficients from admixed and single-continental genomes corresponding to either African or European ancestry. For example, suppose that the coefficient of $M_{ik1}$ is the African-specific Tractor estimate. This coefficient was plotted against the coefficient of $M_{ik}$ obtained from African genomes. Similarly, the coefficient of $M_{ik2}$ was compared to the coefficient of $M_{ik}$ obtained from European genomes. This led to the plots in Fig. 3B and C.

The local ancestry-adjusted LD covariance in Eq. 13 was computed by regressing $C_{ij}$ on $M_{ik1}$, $M_{ik2}$, and $L_{ik2}$ in 10,000 admixed genomes. The coefficient of $M_{ik1}$, corresponding to Africans, was compared to the coefficient of $M_{ik}$ obtained from 5000 African genomes. Similarly, the coefficient of $M_{ik2}$ was compared to the coefficient of $M_{ik}$ obtained from 5000 European genomes. These comparisons are presented in Fig. 4A.

## Supplementary information

Additional file 1. Supplementary figures.

Additional file 2. Supplementary note containing the details of the theoretical argument.

Additional file 3. Peer review history.

### Acknowledgements
We thank Doc Edge (University of Southern California), Carl Veller (University of Chicago), and Graham Coop (University of California, Davis) for providing helpful comments on an earlier version of the manuscript. We exchanged with Jonathan Terhorst (University of Michigan, Ann Arbor) valuable discussions during the revision.

### Review history
The review history is available as Additional file 3.

### Peer review information

### Authors' contributions
H.L. and B.H. conceived the project. H.L. and M.H.L. derived and confirmed the mathematical results. H.L. analyzed the summary statistics, conducted the simulations, and created the graphical figures. K.H. and B.P provided the summary statistics and acquired the permission for their use. B.H. acquired the funding. All authors edited and reviewed the paper. All authors read and approved the work for publication.

### Funding
Hanbin Lee was supported by a grant of the MD-PhD/Medical Scientist Training Program through the Korea Health Industry Development (KHDI), funded by the Ministry of Health and Welfare, Republic of Korea. Moo Hyuk Lee was supported by a grant from the Physician-Scientist Training Program funded by Seoul National University College of Medicine. Buhm Han was supported by the National Research Foundation of Korea (NRF) (Grant number RS-2025-00553579) funded by the Korean government, Ministry of Science, and ICT. Buhm Han was also supported by the Creative-Pioneering Researchers Program funded by Seoul National University and by the AI-Bio Research Grant through Seoul National University.

### Data availability
The codes and scripts used to produce the results of the paper can be found at Zenodo (https://zenodo.org/records/15637591) [65].

## Declarations

### Ethics approval and consent to participate
Not applicable.

### Consent for publication
Not applicable.

### Competing interests
Buhm Han is the CEO of SpintoAI Inc.

### Author details
[1]Present Address: Department of Statistics, University of Michigan, Ann Arbor, MI, USA. [2]Department of Medicine, Seoul National University College of Medicine, Seoul, Republic of Korea. [3]Department of Mathematical Sciences, Seoul National University, Seoul, Republic of Korea. [4]Department of Biomedical Sciences, Seoul National University College of Medicine, Seoul, Republic of Korea. [5]Department of Epidemiology, Harvard T.H. Chan School of Public Health, Boston, MA, USA. [6]Department of Genetics, University of Pennsylvania, Philadelphia, PA, USA. [7]Institute of Biomedical Informatics, University of Pennsylvania, Philadelphia, PA, USA. [8]Interdisciplinary Program in Bioengineering, Seoul National University, Seoul, Republic of Korea. [9]Convergence Dementia Research Center, Seoul National University Medical Research Center, Seoul, Republic of Korea. [10]BK21 Plus Biomedical Science Project, Seoul National University College of Medicine, Seoul, Republic of Korea.

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

## 