## [Additional file 3. Peer review history. · Genome Biology]

Review history

First round of review

Reviewer 1

This paper presents the results of investigations into the performance of different statistical genetics models in admixed populations. This is an important topic, as many researchers struggle to keep up with the different models, and few papers discuss the relative merits of the underlying assumptions. Many of the analyses are novel and (as far as I could verify, but see below) they appear to be correct.

One set of results had to do with expectations and estimator variances for association statistics, while the second part was an empirical comparison of LD-based analyses (specifically LDSC).

I found the first part rather clear and informative: we have some analytical expressions for three estimator variances, and we can use them to understand these estimators better. These are useful results that will be of interest to a moderate number of geneticists.

The take-home message from the second part was less clear. The abstract claims: "Our results show that a mosaic of independent single-continental segments is an insufficient approximation of contemporary admixed populations". But since all models are wrong, it is worth asking: insufficient for what? What has been shown is, much more specifically, that application of LDSC to TRACTOR output gives inconsistent results. The cause of these confusing results is not elucidated (especially, whether it is an issue with the model, or sample size?). Application of LDSC to TRACTOR output is not a common practice (that I know of), so this observation, without delving deeper into its causes, is of moderate interest.

Specific issues:

1) There are at least a few sweeping conclusions that do not seem justified by the results they rely on. E.g.,

1a) The main text and supplementary note argue that confounding is fully addressed by global ancestry correction, stating "further adjustment is not required". If readers follow this recommendation, I fear they may get into trouble. First, all results are derived in a model with no environmental covariates, no assortative mating, no substructure within the source population, etc. etc. So only a very limited type of confounding has been considered. This should be explicit. Even within that limited context, I do not feel like the lack of confounding has been shown. The manuscript only shows that the expected estimated marker effect sizes are proportional to a vector of causal effect sizes (equations 7-10).

I do not agree that this implies a lack of confounding (or at least, it uses a very narrow definition of confounding) What this shows is that there will be no inferred effect sizes if there are no causal effect sizes *anywhere in the genome*. But in that case, the simulated phenotype model is pure gaussian noise. Of course there will be no confounding! I think you can argue a lack of confounding if there is no inferred effect sizes without a *proximal* causal variant. I don't think that this has been shown (or I am missing a step in the argument).

1b) A main section heading is that "the power of the different GWAS methods can be calculated", but then the next paragraph states that powers cannot be compared. As far as I can tell, there is no power calculation performed - only variances of parameter estimates are obtained. Despite this lack of power estimates, there is a claim that admixture mapping will always be more powerful than the Armitage trend test. (Eq24 and line 84).

This seems wrong. If the two admixing populations are statistically identical, for example, admixture mapping should have zero power and the trend test should work just fine. If I am being honest, I don't understand how equation (24) is derived. It is a bit strange that it does not depend on gamma or on beta. Maybe I misunderstand what is meant by "the only-admixture LD scenario" - does that mean that loci are independent within ancestry blocks, so that markers are only correlated with the causal variants by the ancestry blocks? If so, please clarify that it will be more powerful if there is no LD between markers and causal variant!

2)

I found the layout of the paper quite confusing. I don't want to impose my stylistic choices, and authors do not have to follow my recommendations here. But this limited my ability to verify both the math and the conclusions.

a)

Half of the results feel geared at statistical geneticists (analytical proofs!) and the other half at a more applied audience (beware of LDSC from tractor estimates!). I assume that the equations were relegated into a supplementary note to not scare away the second part of the audience. However, the analytical results are the main results from the first part of the paper. The main text is not self-contained without them - it reads as a discussion of the main results, which are not presented. I'm fine with leaving math details in a supplement, but I genuinely could not understand claims made in the Results section before parsing deeply through the supplementary note. So either put the equations in the main text, or write the main text such that it can be understood without reading the supplement.

Examples:

"There is no clear agreement on the form of the parameter". Are we talking about disagreement in notation? Or just that the models measure different things (in which case, there is no expectation that there should be an agreement). I understand that what is meant is the latter, I had no idea until read through the supplement.

L59:

"The covariances between ancestry-specific markers are exactly zero... (equation 18 of Supplementary Note)"? Here again, I had no idea what was meant until I read the supplement. Eq 18 shows that, conditional on local ancestry, the number of alleles received from each ancestry are uncorrelated. But I think what is really meant is "Conditional on the true effect sizes, the estimated population-specific effect sizes are uncorrelated."

"Dropping the marker variables from the regression gives the interpretation of admixture mapping...". Do you simply mean that the tractor model (Equation S3), without the markers, the same as the model of admixture mapping (Equation S4)? If so, what does it add to the argument?

b)

The supplementary note derives results for three distinct models in parallel. As a result, the reader has to infer which equation applies to each model based on subtle distinction between notations (e.g., beta with two subscripts, or bold beta with one subscript is the tractor model, beta with one subscript is Armitage, etc). This would have been a lot easier I think to present them sequentially. I do appreciate that the authors want to draw on the parallels between the models, but then I think that a clearer notation would be helpful.

Specific issues:

Abstract: "can make empirical predictions on GWAS" that is a very vague statement. It would be useful to be explicit about what kind of predictions we are talking about. i.e., variance of parameter estimates.

Figure 3b: Is the overlap significant? Or could the long-range LD be mostly noise? I do see that there is some amount of overlap for very short segments, but it is very hard to say whether this is relevant.

Supplement

P2 L19: "As we show, ... other methods do not"

I don't think that this has been shown.

P2 L26-29:

The sum of Bernoulli is Binomial only if probabilities are equal, and I don't think they are here in general. Do you mean Poisson Binomial? This error came up a few times. Are you assuming that maternal and paternal alleles have the same ancestry?

Typos, and minor issues:

First results page, line 7, missing symbol. Ambiguous subject to "summing".

L18: allele frequenc^{*ies*}, presumably?

L29: "that 'admixture events occurred a few dozens of generations ago:" This is not true of all admixture events!

L 43-45: variance of what? Of the estimated effect size?

Figure 1A: It may help the non-statistical reader to explain why it is important to get an analytical form for the standard error if we can estimate these errors from the data itself.

Figure 1B: Please define acronyms. Assuming FE: Fixed Effects meta-analysis, etc.

Results, p2, L 48: casual - causal

Figure 3a: The scale goes from -.5 to -4.5. So independent of the logartimic basis used (incidentally, which one?) all distances are less than 1 bp. Do you mean Mbp? This is important

to understand the scale of the LD inaccuracies

Supplementary note:

Page 1

Line 5) Is there a fixed number of blocks across individuals?

L10: missing word

Page 2:

"We write $Libl$ to be the ancestral count of ancestry l where $l = 1, \dots, nL$ is the number of ancestries"

I think you mean

"We write $Libl$ to be the ancestral count of ancestry l where $l = 1, \dots, nL$ and n_L is the number of ancestries"

Page 3

P3

Eq(7): Please clarify expectation over what. Assuming this is over the dirichlet-distributed P . I don't think P has been presented as a random variable yet.

L38-39: add (respectively) to clarify? "Has" should be "have". Please use proper and consistent punctuation before equations. Please refer the reader to the part of the text where these equations are derived!

ADM: define as admixture mapping.

P6 line 74.2: missing "in"

P6 L74: "this formula": you mean substituting this formula in eq 17?

Eq 17: I did not see a definition of β^{agg} .

Reviewer 2

This paper presents evidence to support a claim that widely adopted methods for leveraging local ancestry estimates in admixed genomes for GWAS purposes is flawed. Specifically, mathematical theory and results using both simulated and read data are used to show that a mosaic of single-continental segments as an approximation of admixed populations is not sufficient to enable previously accurate estimation of two-locus effects. Some of the results are interesting, however further detail is required to fully support the claims made and to establish a clearer picture of the issue presented. I do think the work is worth pursuing and that a picture is being built of an important issue. However the paper feels incomplete and deserves further work.

Firstly, much of the methodology and analysis is unclear. The mathematical results are relegated to a supporting document which is sparsely and inaccurately referenced. The simulation study is vaguely documented, such that results which depend on the simulation settings cannot be extrapolated. The results provided on the real data feel somewhat cherry-picked and far from adequate.

The notation and terminology is inconsistent with the literature, including publications by some of the authors of this paper. This leads to difficulty in establishing exactly which methods from the literature are being critiqued. Elements of the figures are not discussed in the text.

Finally, the contribution of this paper as stated is not large. Previous related work already established that concordance of effect estimates across admixture components is good for almost all phenotypes for causal markers. This paper focusses on the lack of such concordance for SNPs not deemed to be causal. These are essentially false-positive or even true-negative SNPs; this paper does not discuss how causal SNPs are identified, but presumably most research that attempts to leverage the ePSD model that is critiqued here are interested only in the causal SNPs. Of course these are not typically known but rather inferred or simply tagged by SNPs in high LD with them, however in that case this (and previous) papers show that the ePSD framework performs well.

For example, "Causal effects on complex traits are similar for common variants across segments of different continental ancestries within admixed individuals" advocates focussing on causal effects. This finding is corroborated in this paper, therefore the manuscript can be summarised as finding that non-causal variants effect sizes are not well preserved across ancestries; this is not surprising as the true effect size is zero. i.e. false positive and true-negative effect sizes are not consistently estimated across ancestries.

The commentary paper "Estimation of cross-ancestry genetic correlations within ancestry tracts of admixed samples" by Atkinson summarises the results of the "Causal effects..." paper and states that "The most important finding from this work is that causal effects for common variants were largely similar across ancestries, with height notably bucking the trend and showing a significant admix < 1 ". Again, the key difference between that work and this manuscript is the focus on causal variants, right?

Thus the key finding that "the concordance dropped with the increasing proportion of causal loci" as per Fig 2(c) is already established in the "Causal effects" paper in Fig 6(d,e,f), where the density of causal effects is increased in simulations. This diminishes the novelty of this paper. Similarly, another previous work "On powerful GWAS in admixed populations" showed that "GWAS in admixed populations attain improved power for discovery over homogeneous populations in either scenario - similar or different ancestry-specific allelic effects - thus further supporting the need for larger genomic studies in such populations." Hence this paper's contribution appears small in comparison.

Conversely, "On powerful GWAS in admixed populations" states that "Existing association tests attain increased power over traditional GWAS in admixed populations, even when the causal variant has similar allelic effects across ancestries". It also states that "there is an expected loss of power due to imperfect tagging, although preliminary results suggest that the loss in power is small, particularly when genotype imputation is employed". This appears at odds with the results presented in this paper. If so, a direct repudiation should be included, stating how that analysis differs or is incorrect. Or how the scenario considered differs, whichever is relevant.

Unfortunately, some crucial simulation details are either lacking or unsatisfactory. Only chromosome 22 was simulated; this is very short and within a small number of post-admixture generations very few ancestry switches will be observed. Thus the local ancestry inference will

be very uncertain, as there are simply too few events to train on. What reference panels are used to perform local ancestry inference, or is this taken from the underlying ground truth? The manuscript states that Wright-Fisher was used for 5 generations simulation and then coalescent, but for how many more generations?

The statement in the abstract that "Our results show that a mosaic of independent single-continental segments is an insufficient approximation of contemporary admixed populations" overstates the results presented; the finding relates only to the concordance of effect size estimates.

A more explicit framing of how this paper relates to others in the literature is therefore required. This paper attempts to set out mathematical findings and results based on both simulated and real data to support a claim that existing methods perform poorly. Figure 2 is central to this; there is very low concordance between effect sizes in African versus European ancestries. However, 2(c) confirms results from the literature that concordance is high for causal variants, especially when there are not too many of them. Again, this reduces the contribution of this paper to saying that variants that are not causal do not have consistently estimated effect sizes. How important is this? Should such variants not simply be discarded once determined they are not causal? If establishing which variants are causal is the issue, this should be made clearer and how causal SNPs are chosen in the results presented (e.g. in Figure 2 (c)) do this.

The claim that "comparing LD correlations in admixed and single-continental genomes revealed only moderately concordant patterns" does not seem fully justified. Far apart loci (yellow in Figure 3(a)) have differing levels of LD in admixed versus single-continental genomes. How much is this due to the simulation settings? It's impossible to tell as the specifics of the simulation are not provided. Were both sets of genomes simulated for the same number of generations and population size?

The result that there is a substantial overlap in the distribution of lengths of local ancestries and local ancestry adjusted LD correlations provided in Figure 3(b) is compelling and does lead to concern that inclusion of local ancestry is not sufficient to achieve the aims of the ePSD approach. Perhaps this work can make an important contribution to the literature, but greater care is required to build the case.

Minor Comments:

- What are FE and RE2 in Figure 1(b)? Presumably Fixed and Random Effects models, referred to in the single-line paragraph on Meta Analysis. But no details are provided in the text or figure caption.

- Why were height and BMI selected from the PAGE study for additional attention? Given that height was listed as the only (of 38) phenotypes tested in "Causal effects..." Hou et al to have an r_{admix} statistically significantly lower than 1 and that it is a component of BMI, this feels selective. Were these the most interesting or most extreme of the 19 traits considered?

- Some methodology is presented under Results, namely the PSD and ePSD models. The latter is poorly explained, in fact I originally thought this referred to the Falush et al 2003 extension of

the STRUCTURE model, rather than one in which more accurate estimation of local ancestry, leveraging LD patterns, is applied.

- Check the references to Supplementary Equations:

- Does 18 show that "variances are inversely proportional to the ancestry-specific marker variances similar to standard GWAS

applied to non-admixed genomes" in Page 6 Line 54?

- "Fortunately, a simple analytic expression is deduced for global ancestry adjustment under the PSD model (equation 19 of Supplementary Note)". Is this the correct reference?

- Page 7 Line 3 states that "This method showed improved power over the original Tractor statistics across various quantitative traits (Figure 1b)", however this does not appear to be the case. TRACTOR in orange appears far closer to the 0,1 line. Again, are the "various quantitative traits" here just height and BMI or all 19 mentioned previously?

- "Causal effects..." by Hou et al uses RFMix for local ancestry deconvolution. Is that the method used in this paper for the real data? For the simulated data, is the ground truth used? If not, how is it estimated? What are the panels used in RFMix?

- There is inconsistent terminology and notation with closely related works e.g. $R^2 = r_{\text{admix}}$, ePSD is TRACTOR. How does ePSD relate to the methods presented in "Admix-Kit: an integrated toolkit and pipeline for genetic analyses of admixed populations", which is also written by a team including Hou and Pasaniuc?

Authors' response to reviewers

Reviewer #1

Reviewer: This paper presents the results of investigations into the performance of different statistical genetics models in admixed populations. This is an important topic, as many researchers struggle to keep up with the different models, and few papers discuss the relative merits of the underlying assumptions. Many of the analyses are novel and (as far as I could verify but see below) they appear to be correct.

One set of results had to do with expectations and estimator variances for association statistics, while the second part was an empirical comparison of LD-based analyses (specifically LDSC). I found the first part rather clear and informative: we have some analytical expressions for three estimator variances, and we can use them to understand these estimators better. These are useful results that will be of interest to a moderate number of geneticists.

Response: Thank you for your careful assessment of our paper. We greatly appreciate your efforts on thoroughly going through our arguments. We restructured the manuscript substantially and made the claims more specific. We hope that the changes address your concerns.

Reviewer: The take-home message from the second part was less clear. The abstract claims: "Our results show that a mosaic of independent single-continental segments is an insufficient approximation of contemporary admixed populations". But since all models are wrong, it is worth asking: insufficient for what? What has been shown is, much more specifically, that application of LDSC to TRACTOR output gives inconsistent results. The cause of these confusing results is not elucidated (especially, whether it an issue with the model, or sample size?). Application of LDSC to TRACTOR output is not a common practice (that I know of), so this observation, without delving deeper into its causes, is of moderate interest.

Response: We agree that many parts of the argument were not specific enough. The goal of the LDSC analysis was to evaluate the extended PSD model rather than assessing the appropriateness of LDSC itself. The subsequent simulation-based analyses tried to dissect the results in LDSC analysis further, but we were unsuccessful in explaining the point in our first attempt as reviewers' comments show. We made this more explicit in the following and subsequent sections:

2.6 Testing the extended PSD model in real data and simulations

As **Equation 6** is deduced from the ePSD model, we can indirectly assess the model by testing
**Equation 6**. The equation says that Tractor's ancestry-specific estimates are identical to the sum-
mary statistics had the GWAS conducted on single-continental genomes. Given that the prediction
is correct, we hypothesized that measuring the genetic correlation of Tractor's African and European
effect sizes with the corresponding single-continental summary statistics will produce genetic corre-
lations close to 1 provided the ePSD is correct. Tractor estimates were obtained from the PAGE
cohort and the single-continental summary statistics were obtained from PanUKBB. We used link-
age disequilibrium score regression (LDSC) to estimate the genetic correlation (see **Materials and**
**Methods**).

In 15 quantitative traits, the frequent appearance of negative heritability estimates produced
invalid genetic correlations (**Figure 3A**). Such traits include hemoglobin 1Ac (Hb1Ac), C-reactive
proteins (CRP), diastolic blood pressure (DBP), estimated glomerular filtration rate (eGFR), fasting
glucose, height, platelet count, and waist-to-hip ratio (WHR). In some traits, such as CRP, high-
density cholesterol (HDL), low-density cholesterol (LDL), systolic blood pressure (SBP), and WHR,
genetic correlations of European marginal effect sizes were significantly lower than 1. However, The
confidence intervals were often too wide, especially in African marginal effect sizes, to draw reliable
conclusions.

We conclude that the local LD pattern of a single-continental African segment is different from an admixed African segment based on simulation results.

In sum, what we saw in LDSC analysis of Tractor and within-continental summary statistics is
explained by the differences in local LD patterns between chromosomal segments of the same ancestral
origin from admixed and single-continental genomes. Although the pattern is highly concordant in
very short regions, it differentiates quickly as the region grows in length. Hence, ePSD turns out to
be a good approximation in short regions but performs poorly on longer scales that are as short as
1000kb (or 1cM).

Reviewer: There are at least a few sweeping conclusions that do not seem justified by the results they rely on. e.g., 1a) The main text and supplementary note argue that confounding is fully addressed by global ancestry correction, stating "further adjustment is not required". If readers follow this recommendation, I fear they may get into trouble. First, all results are derived in a model with no environmental covariates, no assortative mating, no substructure within the source population, etc. etc. So only a very limited type of confounding has been considered. This should be explicit. Even within that limited context, I do not feel like the lack of confounding has been shown. The manuscript only shows that the expected estimated marker effect sizes are proportional to a vector of causal effect sizes (equations 7-10). I do not agree that this implies a lack of confounding (or at least, it uses a very narrow definition of confounding). What this shows is that there will be no inferred effect sizes if there are no causal effect sizes *anywhere in the genome*. But in that case, the simulated phenotype model is pure gaussian noise. Of course there will be no confounding! I think you can argue a lack of confounding if there are no inferred effect sizes without a *proximal* causal variant. I don't think that this has been shown (or I am missing a step in the argument).

Response: The reviewer made a fair point, so we made our argument more specific. It is better to reduce the term confounding to genetic confounding due to long-range LD induced by population structure by admixture. As the author pointed out, we implicitly assumed that the source population is homogeneous and there is no assortative mating. We tried to expose our assumptions more explicitly in the Results and Methods. The Supplementary Note also has a line-by-line description of where the assumptions entered the proof.

There are a few assumptions for our result to hold. One is that the covariate \mathbf{X}_i accounts for the
environmental confounding due to the correlation between marker k and ε_i . Another assumption is
that the population structure is solely due to admixture and that the ancestral source populations
are in Hardy-Weinberg equilibrium (HWE). We also rule out important population phenomena such
as assortative mating [37, 38, 39]. We focused on addressing genetic confounding in the context
of admixture and its interplay with various ancestry adjustments. Genetic confounding occurs when
causal variants not tagged by the tested marker affect the trait [19, 23, 40, 41, 36]. This is unavoidable
in the common univariate marginal testing procedures in GWAS. The tested variant can cover only

a small portion of the genome, so the causal variants in the rest of the genome are left behind in the
residuals. These distant causal variants can be correlated with the tested variant due to population
structure, leading to spurious associations in the sense that the marker is picking up signals from
remote regions that are far from its own position. This falls into the category of long-range LD due
to population structure [25, 42].

Reviewer: A main section heading is that "the power of the different GWAS methods can be calculated", but then the next paragraph states that powers cannot be compared. As far as I can tell, there is no power calculation performed - only variances of parameter estimates are obtained. Despite this lack of power estimates, there is a claim that admixture mapping will always be more powerful than the Armitage trend test. (Eq24 and line 84).

This seems wrong. If the two admixing populations are statistically identical, for example, admixture mapping should have zero power and the trend test should work just fine.

If I am being honest, I don't understand how equation (24) is derived. It is a bit strange that it does not depend on gamma or on beta. Maybe I misunderstand what is meant by "the only-admixture LD scenario" - does that mean that loci are independent within ancestry blocks, so that markers are only correlated with the causal variants by the ancestry blocks? If so, please clarify that it will be more powerful if there is no LD between markers and causal variant!

Response: Explicit power calculations were omitted because the effect size of methods is generally different in various ways which makes precise theoretical analysis difficult. However, it is straightforward when the tested variant is causal or very near to the causal variant. We added the following section that analyzes the case:

We highlight that the last matrix is diagonal, which is an important observation that will be repeatedly
 used. This is surprising because it implies that the two regression estimates $\hat{\beta}_{1k}$ and $\hat{\beta}_{2k}$ from the
 same regression **Equation 4** on the same data are independent. We can then see that the statistics

are ordered (in increasing order)

$$\begin{aligned}
 T_{n_I}^{l, \text{Tractor}} / n_I &= \beta_k^2 \cdot 2g_{lk}(1 - g_{lk})\mathbb{E}[P_{il}] \cdot C \\
 &\leq T_{n_I}^{\text{Tractor}} / n_I = \beta_k^2 \cdot 2 \sum_{l=1}^2 g_{lk}(1 - g_{lk})\mathbb{E}[P_{il}] \cdot C \\
 &\leq T_{n_I}^{\text{ATT}} / n_I = \beta_k^2 \cdot \left[2 \sum_{l=1}^2 g_{lk}(1 - g_{lk})\mathbb{E}[P_{il}] + 2(g_{1k} - g_{2k})^2\mathbb{E}[P_{i1}P_{i2}] \right] \cdot C
 \end{aligned} \tag{12}$$

Your speculation that it described the case of no "within-continental" LD between a marker and casual variants is correct. Testing the marker still has some power because standard GWAS can draw signals from admixture LD. However, we dropped the content in this revision that is exclusive to admixture mapping to concentrate on GWAS. Another reason is that we rewrote the derivations without referring to the notion of admixture LD blocks indexed by 'b'.

Reviewer: I found the layout of the paper quite confusing. I don't want to impose my stylistic choices, and authors do not have to follow my recommendations here. But this limited my ability to verify both the math and the conclusions.

Half of the results feel geared at statistical geneticists (analytical proofs!) and the other half at a more applied audience (beware of LDSC from tractor estimates!). I assume that the equations were relegated into a supplementary note to not scare away the second part of the audience. However, the analytical results are the main results from the first part of the paper. The main text is not self-contained without them - it reads as a discussion of the main results, which are not presented. I'm fine with leaving math details in a supplement, but I genuinely could not understand claims made in the Results section before parsing deeply through the supplementary note, so either put the equations in the main text, or write the main text such that it can be understood without reading the supplement.

Response: We restructured the main portion to include all relevant mathematics. The Supplementary Notes now only includes proofs. We think this improved the readability of the manuscript.

Reviewer: "There is no clear agreement on the form of the parameter". Are we talking about disagreement in notation? Or just that the models measure different things (in which case, there is no expectation that there should be an agreement). I understand that what is meant is the latter, I had no idea until read through the supplement.

Response: The clarification of the statement is now stated in:

We can see that the linear equations of the generative model **Equation 2** and the regressions
 **Equation 3 and 4** do not coincide. Hence, the connection between the marker coefficients β_k and
 β_{lk} to the causal effects α_j is obscure. A standard result is that β_k is a linear combination of LD
 parameters and causal effect sizes [34, 35, 36]. For both ATT and Tractor, we can derive equations
 similar to the standard result. Under the generative model of **Equation 2**, we can express the
 coefficients β_k of ATT, β_{lk} , and γ_{lk} of Tractor as a function of allele frequency, LD parameters, and
 α_j . Here, we present the $n_L = 2$ case for exposition. Note that we are assuming the ePSD model
 here.

$$\beta_k = \sum_{j \in [k]} \frac{\mathbb{E}[\sum_l D_{ljk} P_{il}]}{\underbrace{\mathbb{E}[\sum_l g_{lk}(1 - g_{lk})P_{il} + \sum_l g_{lk}^2 P_{il}(1 - P_{il}) - \sum_{l \neq l'} g_{lk} g_{l'k} P_{il} P_{il'}]}_{\text{Within-continental LD}}} \alpha_j + \sum_{j=1}^{n_J} \frac{\sum_{l, l'} g_{lk} f_{l'j} \mathbb{E}[\text{Cov}(L_{ikl}^h, L_{ijl'}^h | \mathbf{P}_i)]}{\underbrace{\mathbb{E}[\sum_l g_{lk}(1 - g_{lk})P_{il} + \sum_l g_{lk}^2 P_{il}(1 - P_{il}) - \sum_{l \neq l'} g_{lk} g_{l'k} P_{il} P_{il'}]}_{\text{Admixture LD}}} \alpha_j \quad (5)$$

$$\beta_{lk} = \sum_{j \in [k]_l} \underbrace{\frac{D_{ljk}}{g_{lk}(1 - g_{lk})}}_{\text{Within-continental LD}} \alpha_j \quad (l = 1, \dots, n_L) \quad (6)$$

$$\gamma_{lk} = \sum_{j \in [k]_l} \left(\frac{f_{lj} - h_{ljk}}{1 - g_{lk}} - \frac{f_{1j} - h_{1jk}}{1 - g_{1k}} \right) \alpha_j \quad (l = 2, \dots, n_L) \quad (7)$$

Reviewer: "The covariances between ancestry-specific markers are exactly zero... (equation 18 of Supplementary Note)"? Here again, I had no idea what was meant until I read the supplement. Eq 18 shows that, conditional on local ancestry, the number of alleles received from each ancestry are uncorrelated. But I think what is really meant is "Conditional on the true effect sizes, the estimated population-specific effect sizes are uncorrelated."

Response: This part is clarified in the Supplementary Note. It means that the sampling distribution of the ancestry-specific estimates is asymptotically independent.

706 A.1 Proof of Equation 11

Equation 11 follows from the following statements (Equation 26 and Equation 27). Note that

Equation 11 assumes the case of two ancestries, i.e., $l = 1, 2$.

$\mathbb{E}[\widetilde{\mathbf{M}}_{ik}^T \widetilde{\mathbf{M}}_{ik}]$ can be calculated as follows:

$$\begin{aligned}
 \mathbb{E}[\widetilde{\mathbf{M}}_{ik}^T \widetilde{\mathbf{M}}_{ik}] &= \text{Var}(\mathbf{M}_{ik} - \mathbb{E}[\mathbf{M}_{ik} | \mathbf{L}_{ik}]) \\
 &= \mathbb{E}[\text{Var}(\mathbf{M}_{ik} | \mathbf{L}_{ik})] \\
 &= 2\mathbb{E}[\text{Var}(\mathbf{M}_{ik}^h | \mathbf{L}_{ik})] \quad \because M_{ik}^m \perp\!\!\!\perp M_{ik}^p \text{ (HWE)} \quad \bullet \\
 &= \begin{bmatrix} 2g_{1k}(1 - g_{1k})\mathbb{E}[P_{i1}] & \cdots & 0 \\ \vdots & \ddots & \vdots \\ 0 & \cdots & 2g_{n_L k}(1 - g_{n_L k})\mathbb{E}[P_{in_L}] \end{bmatrix}
 \end{aligned} \tag{26}$$

Note that only diagonal elements are non-zero.

The remaining term, $\mathbb{E}[\widetilde{M}_{ik}^2]$, is calculated as follows.

$$\begin{aligned}
 \mathbb{E}[\widetilde{M}_{ik}^2] &= \mathbb{E}[\text{Var}(M_{ik} | \mathbf{P}_i)] \\
 &= 2\mathbb{E}[\text{Var}(M_{ik}^h | \mathbf{P}_i)] \quad \because M_{ik}^m \perp\!\!\!\perp M_{ik}^p \text{ (HWE)}
 \end{aligned} \tag{27}$$

Equation 28 at the next subsection finishes the proof. Since we assume the case of two ancestries,

$P_{i1} + P_{i2} = 1$, hence $P_{i1}(1 - P_{i1}) = P_{i2}(1 - P_{i2}) = P_{i1}P_{i2}$, thereby yielding Equation 11

Reviewer: "Dropping the marker variables from the regression gives the interpretation of admixture mapping...". Do you simply mean that the tractor model (Equation S3), without the markers, the same as the model of admixture mapping (Equation S4)? If so, what does it add to the argument?

Response: The initial attempt was to show that the properties of admixture mapping follow as a byproduct of analyzing Tractor. As mentioned earlier, we removed these parts to make the draft clearer.

Reviewer: The supplementary note derives results for three distinct models in parallel. As a result, the reader must infer which equation applies to each model based on subtle distinction between notations (e.g., beta with two subscripts, or bold beta with one subscript is the tractor model, beta with one subscript is Armitage, etc). This would have been a lot easier I think to present them sequentially. I do appreciate that the authors want to draw on the parallels between the models, but then I think that a clearer notation would be helpful.

Response: We extensively restructured the manuscript. Also, we dropped the admixture mapping portion from the manuscript and focused on comparing standard GWAS (ATT) and Tractor. We hope this made an improvement.

Reviewer: Abstract: "can make empirical predictions on GWAS" that is a very vague statement. It would be useful to be explicit about what kind of predictions we are talking about. I.e., variance of parameter estimates.

Response: We changed our abstract to :

Abstract

Admixed populations offer valuable insight into the genetic architecture of complex traits. Many studies have proposed methods for genome-wide association study (GWAS) in admixed populations and various simulation studies have evaluated their performances. In this work, we propose another direction of comparison of recently proposed methods for admixed GWAS from a population genetic viewpoint. Our theoretical approach can mathematically and directly compare the power of methods given that the causal variant is tested. This is done by deriving the variance formula of the methods from the population genetic admixture model. Our results analytically confirm previous observation that the standard GWAS test is more powerful than alternative tests due to leveraging allele frequency heterogeneity in which alternatives do not. As a by-product, we obtain a simple method to improve the power of multi-degrees-of-freedom tests only using summary statistics. We further investigate the problem when the causal variant is not directly known but is detected by tagging variants in linkage disequilibrium (LD). The analysis shows that a genetic segment from admixed genomes may exhibit distinct LD patterns from the single-continental counterpart of the same ancestry.

Reviewer: Figure 3b: Is the overlap significant? Or could the long-range LD be mostly noise? I do see that there is some amount of overlap for very short segments, but it is very hard to say whether this is relevant.

Response: We added the connection of Fig 4B to other parts of the manuscript. Also, we modified the figure to include only the LDs at a shorter distance which is relevant for many downstream methods such as LDSC and fine-mapping (<1cM).

Reviewer: "As we show, ... other methods do not" I don't think that this has been shown.

Response: Yes. This part was omitted in the initial version. We removed the phrase and restructured the manuscript. We agree that it is unclear what "intuitive" means in this context. For some clarification, we added annotations that tell which part of the signal comes from either LD or admixture.

$$\begin{aligned}
\beta_k = & \sum_{j \in [k]} \frac{\mathbb{E}[\sum_l D_{ljk} P_{il}]}{\underbrace{\mathbb{E}[\sum_l g_{lk}(1-g_{lk})P_{il} + \sum_l g_{lk}^2 P_{il}(1-P_{il}) - \sum_{l \neq l'} g_{lk} g_{l'k} P_{il} P_{il'}]}_{\text{Within-continental LD}}} \alpha_j \\
& + \sum_{j=1}^{n_J} \frac{\sum_{l,l'} g_{lk} f_{l'j} \mathbb{E}[\text{Cov}(L_{ikl}^h, L_{ijl'}^h | \mathbf{P}_i)]}{\underbrace{\mathbb{E}[\sum_l g_{lk}(1-g_{lk})P_{il} + \sum_l g_{lk}^2 P_{il}(1-P_{il}) - \sum_{l \neq l'} g_{lk} g_{l'k} P_{il} P_{il'}]}_{\text{Admixture LD}}} \alpha_j
\end{aligned} \tag{5}$$

220

$$\beta_{lk} = \sum_{j \in [k]_l} \frac{D_{ljk}}{g_{lk}(1-g_{lk})} \alpha_j \quad (l = 1, \dots, n_L) \tag{6}$$

$$\gamma_{lk} = \sum_{j \in [k]_l} \left(\frac{f_{lj} - h_{lj}}{1-g_{lk}} - \frac{f_{1j} - h_{1j}}{1-g_{1k}} \right) \alpha_j \quad (l = 2, \dots, n_L) \tag{7}$$

Reviewer: The sum of Bernoulli is Binomial only if probabilities are equal, and I don't think they are here in general. Do you mean Poisson Binomial? This error came up a few times. Are you assuming that maternal and paternal alleles have the same ancestry?

Response: No we do not assume that the parental alleles have the same ancestry. The original description was our mistake in language and the actual derivation did not assume that. We now have updated the description to prevent confusion.

Local ancestry (LA) is assigned according to the probability specified by the global ancestry. At
locus k , local ancestry L_{ikl} counts how many copies of the locus originated from ancestry l . In diploids,

4

including humans, $\sum_{l=1}^{n_L} L_{ikl} = 2$. It follows a multinomial distribution

$$\mathbf{L}_{ik} \mid \mathbf{P}_i \sim \text{Multinomial}(n = 2, p = \mathbf{P}_i) \quad (1)$$

where $\mathbf{L}_{ik} = [L_{ik1}, \dots, L_{ikn_L}]^T$ and $\mathbf{P}_i = [P_{i1}, \dots, P_{in_L}]^T$. Precisely speaking, $\mathbf{L}_{ik}^{\mathbf{h}}$ ($\mathbf{h} = \mathbf{m}$ for
maternal and $\mathbf{h} = \mathbf{p}$ for paternal haplotypes) is sampled from $\mathbf{L}_{ik}^{\mathbf{h}} \mid \mathbf{P}_i \sim \text{Multinomial}(n = 1, p = \mathbf{P}_i)$
and $\mathbf{L}_{ik} = \mathbf{L}_{ik}^{\mathbf{m}} + \mathbf{L}_{ik}^{\mathbf{p}}$. Finally, the genotype $G_{ik}^{\mathbf{h}} = 0, 1$ of haplotype \mathbf{h} is sampled from a Bernoulli
distribution $G_{ik}^{\mathbf{h}} \mid L_{ikl}^{\mathbf{h}} = 1 \sim \text{Bernoulli}(p = f_{lk})$ conditional on $\mathbf{L}_{ik}^{\mathbf{h}}$ where l is the source ancestry of
haplotype h at locus k . f_{lk} is the reference allele frequency at the locus in ancestry l .

Reviewer: First results page, line 7, missing symbol. Ambiguous subject to "summing". L18: allele frequenc*ies*, presumably? L29: "that 'admixture events occurred a few dozens of generations ago:" This is not true of all admixture events! L43-45: variance of what? Of the estimated effect size? Results, p2, L 48: casual - causal

Response: These parts were removed after restructuring the manuscript. The last form of typo has been corrected.

Reviewer: Figure 1A: It may help the non-statistical reader to explain why it is important to get an analytical form for the standard error if we can estimate these errors from the data itself.

Response: We added some details on how admixture reduces the standard error and how that can be shown using the formula.

ATT's (the standard GWAS) test statistics is always larger or equal to the Tractor's statistics. This
advantage is driven by the allele frequency difference $g_{1k} - g_{2k}$ between the two source populations.
This part explains how admixture LD contributes to power. The coefficient being tested remains
the same, but admixture LD increases the test statistics by improving the precision of the estimate,
i.e., a smaller standard error and a narrower confidence interval. The combined statistics $T_{n_l}^{\text{Tractor}}$
additionally suffers from the increased degrees-of-freedom of the null χ^2 -distribution. What's new is
that Tractor is less powerful than standard GWAS even without the effect of increased degrees-of-
freedom because the absolute value of the test statistics is smaller. Also, Tractor does not benefit
from allele frequency heterogeneity across ancestral populations at all, although it still does from
heterogeneous effect sizes. This means that admixed GWAS is unlikely to add more statistical power
than single-continental GWAS if one uses Tractor in the absence of a conspicuous causal effect size
heterogeneity. The gain of ATT from $(g_{1k} - g_{2k})^2 \mathbb{E}[P_{i1}P_{i2}]$ depends on $\mathbb{E}[P_{i1}P_{i2}] = \mathbb{E}[P_{i1}(1 - P_{i1})]$,
which is larger when individuals with equal ancestral proportions from both ancestries are common
in the population (function $f(x) = x(1 - x)$ is maximized at $x = 0.5$). It is important to note that
the power gain driven by $(g_{1k} - g_{2k})^2 \mathbb{E}[P_{i1}P_{i2}]$ is absent in a multi-ancestry cohort only made up of
people of single-continental origins. People from ancestry l will have $P_{il} = 1$ and $P_{il'} = 0$ for $l' \neq l$,
so $\mathbb{E}[P_{il}P_{i2}]$ is always zero.

Reviewer: Figure 1B: Please define acronyms. Assuming FE: Fixed Effects meta-analysis, etc.

Response: The new legend now expands the acronyms.

Figure 2: Predictions of the PSD model evaluated in real data. **A.** Comparison of predicted and estimated standard error of regression coefficients. The top panel is for height and the lower panel is for body-mass index (BMI). **B.** Quantile-Quantile (QQ) plot GWAS results of height (left) and BMI (right) in the PAGE cohort. FE: Fixed-effects meta-analysis, RE2: Han-Eskin random-effects meta-analysis.

Reviewer: Figure 3a: The scale goes from -.5 to -4.5. So independent of the logarithmic basis used (incidentally, which one?) all distances are less than 1 bp. Do you mean Mbp? This is important to understand the scale of the LD inaccuracies

Response: All the scales are now in absolute scale so that they are more readable.

Reviewer #2

Reviewer: This paper presents evidence to support a claim that widely adopted methods for leveraging local ancestry estimates in admixed genomes for GWAS purposes is flawed. Specifically, mathematical theory and results using both simulated and read data are used to show that a mosaic of single-continental segments as an approximation of admixed populations is not sufficient to enable previously accurate estimation of two-locus effects. Some of the results are interesting, however further detail is required to fully support the claims made and to establish a clearer picture of the issue presented. I do think the work is worth pursuing and that a picture is being built of an important issue. However the paper feels incomplete and deserves further work.

Response: We thank the reviewer for carefully reading our paper. We did not intend to make such a strong claim that the method currently being used is flawed. We tried to understand and compare LD patterns of chromosomal segments from the same ancestry in admixed and single-continental genomes, unlike previous works that compared segments from different ancestries in only admixed genomes. We believe that some highly overloaded notation and overlap of authors with previous works have created substantial confusion. We apologize for this mistake and hope that the revised manuscript makes these points clearer. We added an overview figure for this purpose.

Figure 1: Overview of this study. **A.** Power of various GWAS methods applied to different types of cohorts (single-ancestry, multi-ancestry, and admixed) were mathematically compared. **B.** European segments from European genomes were compared to segments of the same ancestry from admixed genomes. The same comparison was made for African segments, too.

Reviewer: Firstly, much of the methodology and analysis is unclear. The mathematical results are relegated to a supporting document which is sparsely and inaccurately referenced. The simulation study is vaguely documented, such that results that depend on the simulation settings cannot be extrapolated. The results provided on the real data feel somewhat cherry-picked and far from adequate.

Response: We restructured and rewrote most parts of the manuscript to prevent tedious back-and-forth between the main section and supplementary materials. We added the following sections to explain the precise simulation setting and software. Selective reporting of real data analysis is mentioned in the other comment. Briefly, we included the results of other traits in the Supplementary Figures 2-7. The following figure is Supplementary Figure 2 and rest of the new figures can be found in the supplementary material.

Supplementary Figure 2. Estimated versus predicted standard error of Tractor and standard GWAS regression coefficients or three quantitative traits. The traits are on the left most of the figure.

Reviewer: The notation and terminology is inconsistent with the literature, including publications by some of the authors of this paper. This leads to difficulty in establishing exactly which methods from the literature are being critiqued. Elements of the figures are not discussed in the text.

Response: We removed heavily overloaded notations (for example, r^2_g). We hope that the restructured draft better aligns with previous literature. We added additional legends and descriptions to the figures. We hope that the new manuscript is sufficiently self-contained for the reader to follow.

Reviewer: Finally, the contribution of this paper as stated is not large. Previous related work already established that concordance of effect estimates across admixture components is good for almost all phenotypes for causal markers. This paper focusses on the lack of such concordance for SNPs not deemed to be causal. These are essentially false-positive or even true-negative SNPs; this paper does not discuss how causal SNPs are identified, but presumably most research that attempts to leverage the ePSD model that is critiqued here are interested only in the causal SNPs. Of course these are not typically known but rather inferred or simply tagged by SNPs in high LD with them, however in that case this (and previous) papers show that the ePSD framework performs well.

For example, "Causal effects on complex traits are similar for common variants across segments of different continental ancestries within admixed individuals" advocates focussing on causal effects. This finding is corroborated in this paper, therefore the manuscript can be summarised as finding that non-causal variants effect sizes are not well preserved across ancestries; this is not surprising as the true effect size is zero. i.e. false positive and true-negative effect sizes are not consistently estimated across ancestries.

Response: The key difference between Hou 2023 paper and this work is that Hou 2023 compares genomic segments from the same admixed individuals from different ancestries, while this paper compares segments from the same ancestry but from different individuals. By doing so, Hou (2023) tried to identify the causal effect size heterogeneity across different ancestry segments, while this paper attempted to identify the LD pattern heterogeneity between the same ancestry segments.

The analysis presented in Hou 2023 uses a new method to compute the genetic correlation of causal effect sizes of different ancestral origins. The method includes all variants, unlike GWAS doing marginal testing. The reason for including all variants are explained in section "Pitfalls of using marginal effect sizes to estimate heterogeneity" and Fig 4 (Hou 2023), which shows different LD patterns in segments of different ancestral origin leads to marginal effect size heterogeneity. Since all variants are included in the model, each variant does not pick up signals from other variants from LD. This allowed the model of Hou 2023 to compare the causal effect without worrying about heterogeneous LD patterns that induce heterogeneity in the marginal effects. As the model does not depend on the LD pattern, the results in the paper have no connection to the validity of ePSD.

This work, on the other hand, shows that the LD patterns, and not the causal effect size, are different in segments of the same ancestral origin. For example, we computed the LD pattern of a variant in admixed genomes that reside on an African segment. Then, we compared this with the LD pattern computed from single-continental African genomes. The same procedure was conducted for European segments.

To repeat, we compared African segments from admixed genomes to African segments from admixed genomes. Hence, unlike the previous work of Hou et al (2023) that compared segments of different ancestries, this work compared segments of the same ancestral origin. Likewise, European segments from admixed genomes were compared to European segments from single-continental genomes. See Figure 1B and Supplementary Figure 1 for a visual comparison.

Figure 1: Overview of this study. **A.** Power of various GWAS methods applied to different types of cohorts (single-ancestry, multi-ancestry, and admixed) were mathematically compared. **B.** European segments from European genomes were compared to segments of the same ancestry from admixed genomes. The same comparison was made for African segments, too.

Supplementary Figure 1. Overview of the analysis of Hou et al. (2023). African and European segments only from admixed genomes were compared to each other.

Reviewer: The commentary paper "Estimation of cross-ancestry genetic correlations within ancestry tracts of admixed samples" by Atkinson summarises the results of the "Causal effects..." paper and states that "The most important finding from this work is that causal effects for common variants were largely similar across ancestries, with height notably bucking the trend and showing a significant admix < 1 ". Again, the key difference between that work and this manuscript is the focus on causal variants, right?

Response: As noted above, we measure the correlation of marginal effects between African segments from admixed and single-continental genomes. We did it, too, for the European segments. Hence, African segments are compared to African segments, and European segments are compared to European segments. After reading one of your comments below, we suspect that our highly overloaded notation caused the confusion. Despite the different goals of our paper and Hou (2023), we used the same notation $r_{g, admix}$ to denote a different quantity.

Reviewer: Thus the key finding that "the concordance dropped with the increasing proportion of causal loci" as per Fig 2(c) is already established in the "Causal effects" paper in Fig 6(d,e,f), where the density of causal effects is increased in simulations. This diminishes the novelty of this paper. Similarly, another previous work "On powerful GWAS in admixed populations" showed that "GWAS in admixed populations attain improved power for discovery over homogeneous populations in either scenario - similar or different ancestry-specific allelic effects - thus further supporting the need for larger genomic studies in such populations." Hence this paper's contribution appears small in comparison.

Response: For the reasons mentioned above, this paper is measuring the concordance between African segments from admixed and single-continental genomes which is comparing segments of the same ancestry (and also for the Europeans). Therefore, it is completely different from comparing African segments to European segments of admixed genomes which compares segments of different ancestry.

Although the power advantage of standard GWAS over Tractor has been repeatedly verified in earlier studies, our analysis explains where the advantage exactly comes from and its precise magnitude in terms of allele frequency parameters. We mathematically show that Tractor does not benefit from heterogeneous allele frequency across populations. Furthermore, we show that admixed individuals confer additional power gain compared to a multi-ancestry cohort consisting of multiple single-continental individuals.

ATT's (the standard GWAS) test statistics is always larger or equal to the Tractor's statistics. This
advantage is driven by the allele frequency difference $g_{1k} - g_{2k}$ between the two source populations.
This part explains how admixture LD contributes to power. The coefficient being tested remains
the same, but admixture LD increases the test statistics by improving the precision of the estimate,
i.e., a smaller standard error and a narrower confidence interval. The combined statistics T_{nl}^{Tractor}
additionally suffers from the increased degrees-of-freedom of the null χ^2 -distribution. What's new is
that Tractor is less powerful than standard GWAS even without the effect of increased degrees-of-
freedom because the absolute value of the test statistics is smaller. Also, Tractor does not benefit
from allele frequency heterogeneity across ancestral populations at all, although it still does from
heterogeneous effect sizes. This means that admixed GWAS is unlikely to add more statistical power
than single-continental GWAS if one uses Tractor in the absence of a conspicuous causal effect size
heterogeneity. The gain of ATT from $(g_{1k} - g_{2k})^2 \mathbb{E}[P_{i1}P_{i2}]$ depends on $\mathbb{E}[P_{i1}P_{i2}] = \mathbb{E}[P_{i1}(1 - P_{i1})]$,
which is larger when individuals with equal ancestral proportions from both ancestries are common
in the population (function $f(x) = x(1 - x)$ is maximized at $x = 0.5$). It is important to note that
the power gain driven by $(g_{1k} - g_{2k})^2 \mathbb{E}[P_{i1}P_{i2}]$ is absent in a multi-ancestry cohort only made up of
people of single-continental origins. People from ancestry l will have $P_{il} = 1$ and $P_{il'} = 0$ for $l' \neq l$,
so $\mathbb{E}[P_{il}P_{i2}]$ is always zero.

Reviewer: Conversely, "On powerful GWAS in admixed populations" states that "Existing association tests attain increased power over traditional GWAS in admixed populations, even when the causal variant has similar allelic effects across ancestries". It also states that "there is an expected loss of power due to imperfect tagging, although preliminary results suggest that the loss in power is small, particularly when genotype imputation is employed". This appears at odds with the results presented in this paper. If so, a direct repudiation should be included, stating how that analysis differs or is incorrect. Or how the scenario considered differs, whichever is relevant.

Response: We clarified how ATT benefits from admixture, unlike Tractor, in response to the previous comment. We believe that we have not claimed on how tagging affects power. In the case of non-causal marker variants, the true marginal effect size differs between methods. Hence, we limited our analysis to the case of testing the causal variant. Overall, our analysis is consistent with previous empirical investigations and adds further theoretical clarification that explain those results.

**2.5 The Pritchard-Stephens-Donnelly model can predict GWAS power**

Standard GWAS (the Armitage trend test, ATT) and Tractor are two popular methods for conducting
GWAS in admixed populations. Simulation-based and empirical comparisons have previously been
made in the literature provided the causal variant was directly tested [44, 45]. We complement the
earlier findings with a precise mathematical formula. In the main text, we present the case of two
source populations. See the **Materials and Methods** for the general result of more than two source
populations. When the tested variant is causal and does not tag any other variants, **Equation 5** and
**6** reduce to

$$\beta_k = \sum_{j=k} \alpha_j = \alpha_k \quad \text{and} \quad \beta_{lk} = \sum_{j=k} \alpha_j = \alpha_k \quad (8)$$

which are identical to the causal effect size of the tested variant.

Reviewer: Unfortunately, some crucial simulation details are either lacking or unsatisfactory. Only chromosome 22 was simulated; this is very short and within a small number of post-admixture generations very few ancestry switches will be observed. Thus the local ancestry inference will be very uncertain, as there are simply too few events to train on. What reference panels are used to perform local ancestry inference, or is this taken from the underlying ground truth? The manuscript states that Wright-Fisher was used for 5 generations simulation and then coalescent, but for how many more generations?

Response: The simulation was poorly described in the original version. We apologize for this. The simulation duration is depicted in Supplementary Figure 8 where the demographic model was adopted from Browning et al. (2018).

Supplementary Figure 8. The demographic model of African-American admixture adopted from Browning et al. (2018). The demes yaml file was retrieved from stdpopsim catalog.

Link: <https://popsim-consortium.github.io/stdpopsim-docs/stable/catalog.html>

We used a longer genome ($\approx 10^8$, roughly the half the size of Chr3), adopting the parameters from chromosome 1. We extracted the local ancestry information from the simulation directly using tskit and tspop, rather than inferring them. Coalescent simulation is retrospective (backwards-in-time), so we cannot fix the number of generations like forward simulations. The simulation lasts until the last coalescence event occurs. In this case, it continues through the pre-out-of-Africa period due to the demographic model where the last coalescent event happens inside Africa. The first five generations of Wright-Fisher refers to turning off the coalescent approximation option in msprime in the first few generations (backwards-in-time) following the suggestion of Nelson et al. (Accounting for long-range correlations in genome-wide simulations of large cohorts).

We added the following:

4.6 African-American genome simulation

The demographic model of African Americans was retrieved from `stdpopsim` 0.2.0 catalog [53]. The
model id we used was `AmericanAdmixture_4B11` that was reported in Browning and colleagues [51].
We excluded the East Asian contribution and modified the admixture proportion of admixed African
Americans to be 80% African and 20% Europeans. We assumed that the admixture occurred 12
generations ago [32, 21, 33]. The resulting demography model file is available in our paper’s github in
`demes` format [66]. A visual presentation of the demographic model is in **Supplementary Figure 8**.

Using the above demography, we simulated 5000, 5000, and 10000 African, European, and African
American individuals with `msprime` 1.3.3 [50, 67]. The genome length was set to 10^8 base pairs
(approximately half the length of chromosome 3). Recombination and mutation rate were set to
1.15×10^{-8} and 1.29×10^{-8} which were adapted from *Homo sapiens* `stdpopsim` catalog [53]. Following
the suggestion from Nelson and colleagues, the first 5 generations (backwards in time) were set to
follow a Wright-Fisher process [57]. For the remaining period in the past, the default coalescent
process of `msprime` was used. Local ancestry information of African Americans were extracted from
the tree sequence generated by `msprime` using `tspop`. By recording all the lineages in the tree sequence

that were present right before the admixture event, `tspop` tracks the ancestry segments of the modern
genome to their ancestors in the ancestral source populations [68].

Note that the number of generations in coalescent simulations is not fixed. It proceeds until all lineages coalesce. Given that the coalescent process is a random stochastic process, the number of generations here is a random variable, rather than a fixed parameter. Under the demographic model in Supplementary Figure 8, the simulation will continue at least until the out-of-Africa event because all lineages should at least the blue bar (African) before them to completely coalesce.

Reviewer: The statement in the abstract that "Our results show that a mosaic of independent single-continental segments is an insufficient approximation of contemporary admixed populations" overstates the results presented; the finding relates only to the concordance of effect size estimates.

Response: The paper shows that given that ePSD is correct, the ancestry-specific estimates of Tractor should be identical to the single-continental counterpart. Since we see a huge discrepancy between the two types of estimates, we provide evidence that ePSD is wrong, which is a proof by contradiction. However, we agree that ePSD cannot be ruled out completely by this single line of evidence.

2.6 Testing the extended PSD model in real data and simulations

As **Equation 6** is deduced from the ePSD model, we can indirectly assess the model by testing
**Equation 6**. The equation says that Tractor's ancestry-specific estimates are identical to the sum-
mary statistics had the GWAS conducted on single-continental genomes. Given that the prediction
is correct, we hypothesized that measuring the genetic correlation of Tractor's African and European
effect sizes with the corresponding single-continental summary statistics will produce genetic corre-
lations close to 1 provided the ePSD is correct. Tractor estimates were obtained from the PAGE
cohort and the single-continental summary statistics were obtained from PanUKBB. We used link-
age disequilibrium score regression (LDSC) to estimate the genetic correlation (see **Materials and**
**Methods**).

Indeed, local ancestry inference methods and, more generally, the coalescent-with-recombination on which those methods are based do not assume that different ancestry blocks are independent. We explain why an independent assumption is frequently adopted in GWAS literature in the following paragraph and beyond:

2.3 The extended PSD and its connection to previous literature

So far, we explained how genotype at a single locus is determined by the PSD model. The model
does not describe the dependence between two or more loci and models the joint distribution as if
the loci are mutually independent [1, 25]. Local ancestry inference methods, whether discriminative
(RFMix) [8] or generative (hidden Markov model-based methods) [7, 10], employ an approximate
coalescent with recombination model to describe the dependence between the loci [25, 26, 27]. In this
framework, LD-related quantities of the ancestral and admixed populations such as LD covariance are
realizations of the aforementioned coalescent process parameterized by the recombination rate and
other evolutionary parameters [28], opposed to most, if not all, studies in GWAS literature that treat
the LD-related quantities as a fixed parameter [29, 30].

Reviewer: A more explicit framing of how this paper relates to others in the literature is therefore required. This paper attempts to set out mathematical findings and results based on both simulated and real data to support a claim that existing methods perform poorly. Figure 2 is central to this; there is very low concordance between effect sizes in African versus European ancestries. However, 2(c) confirms results from the literature that concordance is high for causal variants, especially when there are not too many of them. Again, this reduces the contribution of this paper to saying that variants that are not causal do not have consistently estimated effect sizes. How important is this? Should such variants not simply be discarded once determined they are not causal? If establishing which variants are causal is the issue, this should be made clearer and how causal SNPs are chosen in the results presented (e.g. in Figure 2 (c)) do this.

Response: As stated in the earlier responses, we did not claim that the effect sizes in African versus European ancestries are different. As you've correctly pointed out, this was demonstrated in earlier papers such as Hou 2023. In this work, we compared marginal effect sizes and the LD pattern in African segments from admixed genomes versus African segments from single-continental genomes. We added annotations (highlighted in red) to the figures, as well as descriptions in the manuscript and the methods section.

The local ancestry-adjusted LD covariance in **Equation 13** was computed by regressing C_{ij} on
 M_{ik1} , M_{ik2} , and L_{ik2} in 10000 admixed genomes. The coefficient of M_{ik1} , corresponding to Africans,
 was compared to the coefficient of M_{ik} obtained from 5000 African genomes. Similarly, the coeffi-
 cient of M_{ik2} was compared to the coefficient of M_{ik} obtained from 5000 European genomes. These
 comparisons are presented in **Figure 4A**.

We greatly agree that drawing connections with previous works is essential to position our work in the literature:

2.3 The extended PSD and its connection to previous literature

So far, we explained how genotype at a single locus is determined by the PSD model. The model
does not describe the dependence between two or more loci and models the joint distribution as if
the loci are mutually independent [1, 25]. Local ancestry inference methods, whether discriminative
(RFMix) [8] or generative (hidden Markov model-based methods) [7, 10], employ an approximate
coalescent with recombination model to describe the dependence between the loci [25, 26, 27]. In this
framework, LD-related quantities of the ancestral and admixed populations such as LD covariance are
realizations of the aforementioned coalescent process parameterized by the recombination rate and
other evolutionary parameters [28], opposed to most, if not all, studies in GWAS literature that treat
the LD-related quantities as a fixed parameter [29, 30].

To elaborate, modern admixed genomes are thought of as a realization of the evolutionary process
that began at the time of admixture happened in the past. Coalescent with recombination explicitly
models the recombination process backwards in time [31]. The observed LD patterns in the modern
genome are merely one of the many possibilities that could have materialized from this random process
[28]. Note that current allele frequencies of the source and admixed populations are also realizations
of the random process in this setting due to genetic drift and mutation [7, 25]. In GWAS literature,
we implicitly condition the current state of the population and treat the contemporary LD patterns
as a parameter of the current population [29, 30]. For instance, the famous linkage disequilibrium
score regression (LDSC) uses the sample LD scores estimated from the reference panel to approximate
the population LD correlation of contemporary populations. It does not make any reference to the
underlying evolutionary process of recombination. Only the realizations of the process as a collection
of LD-related parameters are considered.

By changing Figure 4A, we provide a clearer picture at what distance the ePSD model start to fail significantly. The range is between 0.1 to 1cM (or 100kb to 1000kb) which is within a range that is considered local in the literature. For example, LD scores are usually computed in a 1cM window. In this range of window, causal variants cannot be identified without the help of fine-mapping techniques, and statistical fine-mapping requires correct LD information around the lead variant. Therefore, discordance of marginal effect size is a practically important issue.

A

We repeatedly highlight that the x-axis is the LD covariance in African (or European) segments computed from admixed genomes and the y-axis is the LD covariance in African (or European) segments computed from single-continental genomes. African LDs are compared to each other and European LD are compared to each other. We never compared African LD to European LD in this work. See Figure 1B and Supplementary Figure 1 for a visual comparison of the two studies.

Reviewer: The claim that "comparing LD correlations in admixed and single-continental genomes revealed only moderately concordant patterns" does not seem fully justified. Far apart loci (yellow in Figure 3(a)) have differing levels of LD in admixed versus single-continental genomes. How much is this due to the simulation settings? It's impossible to tell as the specifics of the simulation are not provided. Were both sets of genomes simulated for the same number of generations and population size?

Response: We apologize for the poor description of the simulation settings. The method section now contains the details. Briefly, we split the window size to make the figure more readable and discarded too distant pairs of variants (Fig 4A). The simulation has been conducted as mentioned in the earlier responses. All the individuals were collected from the same run of simulation that follows a demographic model of modern American population starting from Africa in the past (Supplementary Figure 8).

Reviewer: The result that there is a substantial overlap in the distribution of lengths of local ancestries and local ancestry adjusted LD correlations provided in Figure 3(b) is compelling and does lead to concern that inclusion of local ancestry is not sufficient to achieve the aims of the

ePSD approach. Perhaps this work can make an important contribution to the literature, but greater care is required to build the case.

Response: We have written the following section to draw connection between LDSC analysis and the subsequent simulation analyses.

Indeed, comparing normalized LD covariances (**Equation 13 and 14**) of admixed and single-
continental genomes revealed only moderately concordant patterns. Here, we compared African
(European) segments from admixed individuals to the corresponding segments in single-continental
(non-admixed) African (European) genomes, which is different from a previous study that compared
segments of different ancestries (African versus Europeans) all obtained from admixed individuals
24. We binned the pairs of loci according to their physical distance (**Figure 4A**). The correlation
was fairly high within the range ($\sim 0.1\text{cM}$ or 100kb) of within-continental LD blocks 54 55, but con-
siderably low in the ranges ($\sim 1\text{cM}$ or 1000kb) considered in LD scores 29. The correlation's decay
were faster in European ancestry that had lower occupancy in the genome (African:European = 8 : 2).
This is likely because the LA segments from the minority population is more easily surrounded by the
majority LA segments.

Reviewer: What are FE and RE2 in Figure 1(b)? Presumably Fixed and Random Effects models, referred to the single-line paragraph on Meta Analysis. But no details are provided in the text or figure caption.

Response: We added the following descriptions:

The ancestry-specific test statistics of Tractor $T_{n_I}^{l, \text{Tractor}}$ are mutually independent for all pairs
of $l = 1, \dots, n_L$ (**Equation 11** and **Materials and Methods**). This is surprising because all
the ancestry-specific statistics are obtained from the same regression and data. The independence
allows us to combine them using existing meta-analysis methods that combine independent summary
statistics. The widely-adopted fixed-effects (FE) meta-analysis produces a test statistics equal to
the Tractor’s combined test [46, 47]. Despite having the same test statistics value, it only has one
degrees-of-freedom because it tests a different hypothesis. FE meta-analysis tests a single-parameter
hypothesis ($H_0 : \beta_k = 0$) assuming that all ancestry-specific effect sizes are equal, and Tractor tests
a n_L -parameter hypothesis that allows all effect sizes to vary ($H_0 : \beta_{lk} = 0$ for all $l = 1, \dots, n_L$).

10

We applied another meta-analysis based on the Han-Eskin model, commonly abbreviated as RE2
(random-effects two) [48, 49]. When applied to the same PAGE data of height and BMI, we found
that FE and RE2 achieve better power than Tractor’s combined test (**Figure 2B**). RE2 has a 1.5
degrees-of-freedom and is known to be powerful when the effect size estimates are heterogeneous.
The fact that it performs worse than ATT and similarly to FE suggests that the marginal effect size
heterogeneity is not large in the two traits. The results remain the same in other quantitative traits
as well (**Supplementary Figure 6-7**).

Figure 2: Predictions of the PSD model evaluated in real data. **A.** Comparison of predicted and estimated standard error of regression coefficients. The top panel is for height and the lower panel is for body-mass index (BMI). **B.** Quantile-Quantile (QQ) plot GWAS results of height (left) and BMI (right) in the PAGE cohort. FE: Fixed-effects meta-analysis, RE2: Han-Eskin random-effects meta-analysis.

Additional details can be found in the method section.

Reviewer: Why were height and BMI selected from the PAGE study for additional attention? Given that height was listed as the only (of 38) phenotypes tested in "Causal effects..." Hou et al to have an r_{admix} statistically significantly lower than 1 and that it is a component of BMI, this feels selective. Were these the most interesting or most extreme of the 19 traits considered?

Response: The new supplementary figures 2-8 now contains other 12 traits that were available in the PAGE cohort. The initial choice was simply because height and BMI were those with the largest sample sizes (non-missing measurements).

Reviewer: Some methodology is presented under Results, namely the PSD and ePSD models. The latter is poorly explained, in fact I originally thought this referred to the Falush et al 2003 extension of the STRUCTURE model, rather than one in which more accurate estimation of local ancestry, leveraging LD patterns, is applied.

Response: Thank you for pointing out an important reference. The following paragraph and the subsequent ones now include how ePSD and other models are related.

2.3 The extended PSD and its connection to previous literature

So far, we explained how genotype at a single locus is determined by the PSD model. The model
does not describe the dependence between two or more loci and models the joint distribution as if
the loci are mutually independent [1, 25]. Local ancestry inference methods, whether discriminative
(RFMix) [8] or generative (hidden Markov model-based methods) [7, 10], employ an approximate
coalescent with recombination model to describe the dependence between the loci [25, 26, 27]. In this
framework, LD-related quantities of the ancestral and admixed populations such as LD covariance are
realizations of the aforementioned coalescent process parameterized by the recombination rate and
other evolutionary parameters [28], opposed to most, if not all, studies in GWAS literature that treat
the LD-related quantities as a fixed parameter [29, 30].

To elaborate, modern admixed genomes are thought of as a realization of the evolutionary process
that began at the time of admixture happened in the past. Coalescent with recombination explicitly
models the recombination process backwards in time [31]. The observed LD patterns in the modern
genome are merely one of the many possibilities that could have materialized from this random process
[28]. Note that current allele frequencies of the source and admixed populations are also realizations
of the random process in this setting due to genetic drift and mutation [7, 25]. In GWAS literature,
we implicitly condition the current state of the population and treat the contemporary LD patterns
as a parameter of the current population [29, 30]. For instance, the famous linkage disequilibrium
score regression (LDSC) uses the sample LD scores estimated from the reference panel to approximate
the population LD correlation of contemporary populations. It does not make any reference to the
underlying evolutionary process of recombination. Only the realizations of the process as a collection
of LD-related parameters are considered.

Reviewer: Check the references to Supplementary Equations:

Response: We have rewritten the supplementary note and believe that the references are now correct.

Reviewer: Does 18 show that "variances are inversely proportional to the ancestry-specific marker variances similar to standard GWAS applied to non-admixed genomes" in Page 6 Line 54?

Response: It must be seen in conjunction with equation 17 to get the conclusion. In the new version, see equation 9 and 20.

Reviewer: "Fortunately, a simple analytic expression is deduced for global ancestry adjustment under the PSD model (equation 19 of Supplementary Note)". Is this the correct reference?

Response: It is the correct reference, but we admit that it looks very obscure. In the new version, please see equation 9.

Reviewer: Page 7 Line 3 states that "This method showed improved power over the original Tractor statistics across various quantitative traits (Figure 1b)", however this does not appear to be the case. TRACTOR in orange appears far closer to the 0,1 line. Again, are the "various quantitative traits" here just height and BMI or all 19 mentioned previously?

Response: A method is less powerful if the points are closer to the main diagonal. You can see that the blue line (ATT=Standard GWAS) has smaller P-values, followed by meta-analysis approaches. The original Tractor test is the least powerful. We included all the remaining traits in PAGE cohort in the supplementary figures. All traits follow the same pattern found in height and BMI.

Reviewer: "Causal effects..." by Hou et al uses RFMix for local ancestry deconvolution. Is that the method used in this paper for the real data? For the simulated data, is the ground truth used? If not, how is it estimated? What are the panels used in RFMix?

Response:

Pre-GWAS steps, including local ancestry inference using RFMix, of the real genotype data were identical to the Hou et al. study. We added the following paragraph:

**4.1 PAGE and UK Biobank summary statistics**
We analyzed summary statistics of one dataset of African-European admixed individuals. Population
Architecture through Genomics and Environment (PAGE) study included 17,299 genotyped individ-
uals with African-European admixed ancestries determined by estimated admixture proportion and
with approximately 6.9 million variants. Detailed steps of quality control and processing, including
GWAS using `admix-kit` and local ancestry inference using RFMix, can be found in **Genotype data**
**processing** section of Hou and colleagues [21]. Summary statistics of 15 traits of UK Biobank par-
ticipants of African and European ancestry were downloaded from the Pan UK Biobank repository
(<https://pan.ukbb.broadinstitute.org/>).

For simulated data, the local ancestry tracts were ground-truth. The software and algorithms are described in the method section.

**4.6 African-American genome simulation**
The demographic model of African Americans was retrieved from `stdpopsim` 0.2.0 catalog [53]. The
model id we used was `AmericanAdmixture_4B11` that was reported in Browning and colleagues [51].
We excluded the East Asian contribution and modified the admixture proportion of admixed African
Americans to be 80% African and 20% Europeans. We assumed that the admixture occurred 12
generations ago [32, 21, 33]. The resulting demography model file is available in our paper's github in
`demes` format [66]. A visual presentation of the demographic model is in **Supplementary Figure 8**.
Using the above demography, we simulated 5000, 5000, and 10000 African, European, and African
American individuals with `msprime` 1.3.3 [50, 67]. The genome length was set to 10^8 base pairs
(approximately half the length of chromosome 3). Recombination and mutation rate were set to
1.15×10^{-8} and 1.29×10^{-8} which were adapted from *Homo sapiens* `stdpopsim` catalog [53]. Following
the suggestion from Nelson and colleagues, the first 5 generations (backwards in time) were set to
follow a Wright-Fisher process [57]. For the remaining period in the past, the default coalescent
process of `msprime` was used. Local ancestry information of African Americans were extracted from
the tree sequence generated by `msprime` using `tspop`. By recording all the lineages in the tree sequence

that were present right before the admixture event, `tspop` tracks the ancestry segments of the modern
genome to their ancestors in the ancestral source populations [68].

Reviewer: There is inconsistent terminology and notation with closely related works e.g. $R^2 = r_{\text{admix}}$, ePSD is TRACTOR. How does ePSD relate to the methods presented in "Admix-Kit: an integrated toolkit and pipeline for genetic analyses of admixed populations", which is also written by a team including Hou and Pasaniuc?

Response: Admix-Kit uses HAPGEN2 and Haptools2 which are based on the Li-Stephens model. Li-Stephens model is an approximation of the coalescent-with-recombination process that fully describes the neutral evolution of the genome under recombination and genetic drift. These models explicitly model the recombination process. Roughly speaking, ePSD is an even coarse approximation of them that does not explicitly take the recombination process into account.

We elaborate on this point in the following and the subsequent paragraphs:

**2.3 The extended PSD and its connection to previous literature**

So far, we explained how genotype at a single locus is determined by the PSD model. The model
does not describe the dependence between two or more loci and models the joint distribution as if
the loci are mutually independent [1, 25]. Local ancestry inference methods, whether discriminative
(RFMix) [8] or generative (hidden Markov model-based methods) [7, 10], employ an approximate
coalescent with recombination model to describe the dependence between the loci [25, 26, 27]. In this
framework, LD-related quantities of the ancestral and admixed populations such as LD covariance are
realizations of the aforementioned coalescent process parameterized by the recombination rate and
other evolutionary parameters [28], opposed to most, if not all, studies in GWAS literature that treat
the LD-related quantities as a fixed parameter [29, 30].

To elaborate, modern admixed genomes are thought of as a realization of the evolutionary process
that began at the time of admixture happened in the past. Coalescent with recombination explicitly
models the recombination process backwards in time [31]. The observed LD patterns in the modern
genome are merely one of the many possibilities that could have materialized from this random process
[28]. Note that current allele frequencies of the source and admixed populations are also realizations
of the random process in this setting due to genetic drift and mutation [7, 25]. In GWAS literature,
we implicitly condition the current state of the population and treat the contemporary LD patterns
as a parameter of the current population [29, 30]. For instance, the famous linkage disequilibrium
score regression (LDSC) uses the sample LD scores estimated from the reference panel to approximate
the population LD correlation of contemporary populations. It does not make any reference to the
underlying evolutionary process of recombination. Only the realizations of the process as a collection
of LD-related parameters are considered.

Second round of review

Reviewer 2

Overall the manuscript has been vastly improved with a clearer focus and should be of interest to anyone working on GWAS and / or admixture.

In the first review I incorrectly interpreted the contribution claimed in the paper. The authors response and updated manuscript clarified this and I fully accept that a large part of my initial criticisms do not apply to this paper. Specifically, this paper does not compare effect size estimates from segments of admixed genomes inferred to be from differing ancestries; rather it compares effect size estimates arising from admixed genomes' segments inferred to be from a particular ancestry to estimates arising from single-continental genomes of that same ancestry. However, as the other reviewer pointed out, application of LDSC to TRACTOR output is not something that is routinely done. The updated manuscript therefore expands upon the simulation study (which lacked much detail in the initial manuscript) in order to determine why the two estimates differ. Local LD patterns caused by admixture, but unsuccessfully disentangled by local ancestry inference are found to be the key issue, as explored via the simulation study. Perhaps the authors agree that my misinterpretation could befall other readers, as the inclusion of the new Figure 1 serves to clarify the novelty of this paper in contrast to existing literature on GWAS in admixed genomes.

The initial submission was poorly structured and this revision has therefore moved material from the appendices to the main text and removed some unimportant distractions. The notation has been carefully amended to help avoid the type of misinterpretation that I made above and several sections have been entirely rewritten. The result is a much improved manuscript with a clear contribution that is both interesting and important. In short, the authors derive theoretical results that predict the moments of GWAS (ATT), TRACTOR (ePSD), and related extensions based estimators of effect sizes based on admixed genomes. They show that admixed LD differs from single-continental LD and this is the reason behind effect size low concordance for non-causal markers. The issue arises due to the invalidity of the ePSD model's key assumption that within-continental LD cannot stretch beyond local ancestry segments.

Suggestions

1. I tentatively suggest consideration of the paper's title. It does not suggest what the specific contributions of the paper actually are.
2. The legend for the scatter plot in Fig 1A is too small to read. Similarly for Fig 2B.
3. γ in Equation 4 is not defined or discussed in the text (Eq 7 provides the mathematical definition).

Authors' response to reviewers

Reviewer #1

Reviewer: This paper presents the results of investigations into the performance of different statistical genetics models in admixed populations. This is an important topic, as many researchers struggle to keep up with the different models, and few papers discuss the relative merits of the underlying assumptions. Many of the analyses are novel and (as far as I could verify but see below) they appear to be correct.

One set of results had to do with expectations and estimator variances for association statistics, while the second part was an empirical comparison of LD-based analyses (specifically LDSC). I found the first part rather clear and informative: we have some analytical expressions for three estimator variances, and we can use them to understand these estimators better. These are useful results that will be of interest to a moderate number of geneticists.

Response: Thank you for your careful assessment of our paper. We greatly appreciate your efforts on thoroughly going through our arguments. We restructured the manuscript substantially and made the claims more specific. We hope that the changes address your concerns.

Reviewer: The take-home message from the second part was less clear. The abstract claims: "Our results show that a mosaic of independent single-continental segments is an insufficient approximation of contemporary admixed populations". But since all models are wrong, it is worth asking: insufficient for what? What has been shown is, much more specifically, that application of LDSC to TRACTOR output gives inconsistent results. The cause of these confusing results is not elucidated (especially, whether it an issue with the model, or sample size?). Application of LDSC to TRACTOR output is not a common practice (that I know of), so this observation, without delving deeper into its causes, is of moderate interest.

Response: We agree that many parts of the argument were not specific enough. The goal of the LDSC analysis was to evaluate the extended PSD model rather than assessing the appropriateness of LDSC itself. The subsequent simulation-based analyses tried to dissect the results in LDSC analysis further, but we were unsuccessful in explaining the point in our first attempt as reviewers' comments show. We made this more explicit in the following and subsequent sections:

2.6 Testing the extended PSD model in real data and simulations

As **Equation 6** is deduced from the ePSD model, we can indirectly assess the model by testing
**Equation 6**. The equation says that Tractor's ancestry-specific estimates are identical to the sum-
mary statistics had the GWAS conducted on single-continental genomes. Given that the prediction
is correct, we hypothesized that measuring the genetic correlation of Tractor's African and European
effect sizes with the corresponding single-continental summary statistics will produce genetic corre-
lations close to 1 provided the ePSD is correct. Tractor estimates were obtained from the PAGE
cohort and the single-continental summary statistics were obtained from PanUKBB. We used link-
age disequilibrium score regression (LDSC) to estimate the genetic correlation (see **Materials and**
**Methods**).

In 15 quantitative traits, the frequent appearance of negative heritability estimates produced
invalid genetic correlations (**Figure 3A**). Such traits include hemoglobin 1Ac (Hb1Ac), C-reactive
proteins (CRP), diastolic blood pressure (DBP), estimated glomerular filtration rate (eGFR), fasting
glucose, height, platelet count, and waist-to-hip ratio (WHR). In some traits, such as CRP, high-
density cholesterol (HDL), low-density cholesterol (LDL), systolic blood pressure (SBP), and WHR,
genetic correlations of European marginal effect sizes were significantly lower than 1. However, The
confidence intervals were often too wide, especially in African marginal effect sizes, to draw reliable
conclusions.

We conclude that the local LD pattern of a single-continental African segment is different from an admixed African segment based on simulation results.

In sum, what we saw in LDSC analysis of Tractor and within-continental summary statistics is
explained by the differences in local LD patterns between chromosomal segments of the same ancestral
origin from admixed and single-continental genomes. Although the pattern is highly concordant in
very short regions, it differentiates quickly as the region grows in length. Hence, ePSD turns out to
be a good approximation in short regions but performs poorly on longer scales that are as short as
1000kb (or 1cM).

Reviewer: There are at least a few sweeping conclusions that do not seem justified by the results they rely on. e.g., 1a) The main text and supplementary note argue that confounding is fully addressed by global ancestry correction, stating "further adjustment is not required". If readers follow this recommendation, I fear they may get into trouble. First, all results are derived in a model with no environmental covariates, no assortative mating, no substructure within the source population, etc. etc. So only a very limited type of confounding has been considered. This should be explicit. Even within that limited context, I do not feel like the lack of confounding has been shown. The manuscript only shows that the expected estimated marker effect sizes are proportional to a vector of causal effect sizes (equations 7-10). I do not agree that this implies a lack of confounding (or at least, it uses a very narrow definition of confounding). What this shows is that there will be no inferred effect sizes if there are no causal effect sizes *anywhere in the genome*. But in that case, the simulated phenotype model is pure gaussian noise. Of course there will be no confounding! I think you can argue a lack of confounding if there are no inferred effect sizes without a *proximal* causal variant. I don't think that this has been shown (or I am missing a step in the argument).

Response: The reviewer made a fair point, so we made our argument more specific. It is better to reduce the term confounding to genetic confounding due to long-range LD induced by population structure by admixture. As the author pointed out, we implicitly assumed that the source population is homogeneous and there is no assortative mating. We tried to expose our assumptions more explicitly in the Results and Methods. The Supplementary Note also has a line-by-line description of where the assumptions entered the proof.

There are a few assumptions for our result to hold. One is that the covariate \mathbf{X}_i accounts for the
environmental confounding due to the correlation between marker k and ε_i . Another assumption is
that the population structure is solely due to admixture and that the ancestral source populations
are in Hardy-Weinberg equilibrium (HWE). We also rule out important population phenomena such
as assortative mating [37, 38, 39]. We focused on addressing genetic confounding in the context
of admixture and its interplay with various ancestry adjustments. Genetic confounding occurs when
causal variants not tagged by the tested marker affect the trait [19, 23, 40, 41, 36]. This is unavoidable
in the common univariate marginal testing procedures in GWAS. The tested variant can cover only

a small portion of the genome, so the causal variants in the rest of the genome are left behind in the
residuals. These distant causal variants can be correlated with the tested variant due to population
structure, leading to spurious associations in the sense that the marker is picking up signals from
remote regions that are far from its own position. This falls into the category of long-range LD due
to population structure [25, 42].

Reviewer: A main section heading is that "the power of the different GWAS methods can be calculated", but then the next paragraph states that powers cannot be compared. As far as I can tell, there is no power calculation performed - only variances of parameter estimates are obtained. Despite this lack of power estimates, there is a claim that admixture mapping will always be more powerful than the Armitage trend test. (Eq24 and line 84).

This seems wrong. If the two admixing populations are statistically identical, for example, admixture mapping should have zero power and the trend test should work just fine.

If I am being honest, I don't understand how equation (24) is derived. It is a bit strange that it does not depend on gamma or on beta. Maybe I misunderstand what is meant by "the only-admixture LD scenario" - does that mean that loci are independent within ancestry blocks, so that markers are only correlated with the causal variants by the ancestry blocks? If so, please clarify that it will be more powerful if there is no LD between markers and causal variant!

Response: Explicit power calculations were omitted because the effect size of methods is generally different in various ways which makes precise theoretical analysis difficult. However, it is straightforward when the tested variant is causal or very near to the causal variant. We added the following section that analyzes the case:

283 We highlight that the last matrix is diagonal, which is an important observation that will be repeatedly
 used. This is surprising because it implies that the two regression estimates $\hat{\beta}_{1k}$ and $\hat{\beta}_{2k}$ from the
 same regression **Equation 4** on the same data are independent. We can then see that the statistics

are ordered (in increasing order)

$$\begin{aligned}
 T_{n_I}^{l, \text{Tractor}} / n_I &= \beta_k^2 \cdot 2g_{lk}(1 - g_{lk})\mathbb{E}[P_{il}] \cdot C \\
 &\leq T_{n_I}^{\text{Tractor}} / n_I = \beta_k^2 \cdot 2 \sum_{l=1}^2 g_{lk}(1 - g_{lk})\mathbb{E}[P_{il}] \cdot C \\
 &\leq T_{n_I}^{\text{ATT}} / n_I = \beta_k^2 \cdot \left[2 \sum_{l=1}^2 g_{lk}(1 - g_{lk})\mathbb{E}[P_{il}] + 2(g_{1k} - g_{2k})^2 \mathbb{E}[P_{i1}P_{i2}] \right] \cdot C
 \end{aligned} \tag{12}$$

Your speculation that it described the case of no "within-continental" LD between a marker and casual variants is correct. Testing the marker still has some power because standard GWAS can draw signals from admixture LD. However, we dropped the content in this revision that is exclusive to admixture mapping to concentrate on GWAS. Another reason is that we rewrote the derivations without referring to the notion of admixture LD blocks indexed by 'b'.

Reviewer: I found the layout of the paper quite confusing. I don't want to impose my stylistic choices, and authors do not have to follow my recommendations here. But this limited my ability to verify both the math and the conclusions.

Half of the results feel geared at statistical geneticists (analytical proofs!) and the other half at a more applied audience (beware of LDSC from tractor estimates!). I assume that the equations were relegated into a supplementary note to not scare away the second part of the audience. However, the analytical results are the main results from the first part of the paper. The main text is not self-contained without them - it reads as a discussion of the main results, which are not presented. I'm fine with leaving math details in a supplement, but I genuinely could not understand claims made in the Results section before parsing deeply through the supplementary note, so either put the equations in the main text, or write the main text such that it can be understood without reading the supplement.

Response: We restructured the main portion to include all relevant mathematics. The Supplementary Notes now only includes proofs. We think this improved the readability of the manuscript.

Reviewer: "There is no clear agreement on the form of the parameter". Are we talking about disagreement in notation? Or just that the models measure different things (in which case, there is no expectation that there should be an agreement). I understand that what is meant is the latter, I had no idea until read through the supplement.

Response: The clarification of the statement is now stated in:

We can see that the linear equations of the generative model **Equation 2** and the regressions
 **Equation 3 and 4** do not coincide. Hence, the connection between the marker coefficients β_k and
 β_{lk} to the causal effects α_j is obscure. A standard result is that β_k is a linear combination of LD
 parameters and causal effect sizes [34, 35, 36]. For both ATT and Tractor, we can derive equations
 similar to the standard result. Under the generative model of **Equation 2**, we can express the
 coefficients β_k of ATT, β_{lk} , and γ_{lk} of Tractor as a function of allele frequency, LD parameters, and
 α_j . Here, we present the $n_L = 2$ case for exposition. Note that we are assuming the ePSD model
 here.

$$\beta_k = \underbrace{\sum_{j \in [k]} \frac{\mathbb{E}[\sum_l D_{ljk} P_{il}]}{\mathbb{E}[\sum_l g_{lk}(1-g_{lk})P_{il} + \sum_l g_{lk}^2 P_{il}(1-P_{il}) - \sum_{l \neq l'} g_{lk} g_{l'k} P_{il} P_{il'}]}}_{\text{Within-continental LD}} \alpha_j + \underbrace{\sum_{j=1}^{n_J} \frac{\sum_{l,l'} g_{lk} f_{l'j} \mathbb{E}[\text{Cov}(L_{ikl}^h, L_{ijl'}^h | \mathbf{P}_i)]}{\mathbb{E}[\sum_l g_{lk}(1-g_{lk})P_{il} + \sum_l g_{lk}^2 P_{il}(1-P_{il}) - \sum_{l \neq l'} g_{lk} g_{l'k} P_{il} P_{il'}]}}_{\text{Admixture LD}} \alpha_j \quad (5)$$

$$\beta_{lk} = \underbrace{\sum_{j \in [k]_l} \frac{D_{ljk}}{g_{lk}(1-g_{lk})}}_{\text{Within-continental LD}} \alpha_j \quad (l = 1, \dots, n_L) \quad (6)$$

$$\gamma_{lk} = \sum_{j \in [k]_l} \left(\frac{f_{lj} - h_{ljk}}{1-g_{lk}} - \frac{f_{1j} - h_{1jk}}{1-g_{1k}} \right) \alpha_j \quad (l = 2, \dots, n_L) \quad (7)$$

Reviewer: "The covariances between ancestry-specific markers are exactly zero... (equation 18 of Supplementary Note)"? Here again, I had no idea what was meant until I read the supplement. Eq 18 shows that, conditional on local ancestry, the number of alleles received from each ancestry are uncorrelated. But I think what is really meant is "Conditional on the true effect sizes, the estimated population-specific effect sizes are uncorrelated."

Response: This part is clarified in the Supplementary Note. It means that the sampling distribution of the ancestry-specific estimates is asymptotically independent.

706 A.1 Proof of Equation 11

Equation 11 follows from the following statements (Equation 26 and Equation 27). Note that

Equation 11 assumes the case of two ancestries, i.e., $l = 1, 2$.

$\mathbb{E}[\widetilde{\mathbf{M}}_{ik}^T \widetilde{\mathbf{M}}_{ik}]$ can be calculated as follows:

$$\begin{aligned}
 \mathbb{E}[\widetilde{\mathbf{M}}_{ik}^T \widetilde{\mathbf{M}}_{ik}] &= \text{Var}(\mathbf{M}_{ik} - \mathbb{E}[\mathbf{M}_{ik} | \mathbf{L}_{ik}]) \\
 &= \mathbb{E}[\text{Var}(\mathbf{M}_{ik} | \mathbf{L}_{ik})] \\
 &= 2\mathbb{E}[\text{Var}(\mathbf{M}_{ik}^h | \mathbf{L}_{ik})] \quad \because M_{ik}^m \perp\!\!\!\perp M_{ik}^p \text{ (HWE)} \quad \bullet \\
 &= \begin{bmatrix} 2g_{1k}(1 - g_{1k})\mathbb{E}[P_{i1}] & \cdots & 0 \\ \vdots & \ddots & \vdots \\ 0 & \cdots & 2g_{n_L k}(1 - g_{n_L k})\mathbb{E}[P_{in_L}] \end{bmatrix}
 \end{aligned} \tag{26}$$

Note that only diagonal elements are non-zero.

The remaining term, $\mathbb{E}[\widetilde{M}_{ik}^2]$, is calculated as follows.

$$\begin{aligned}
 \mathbb{E}[\widetilde{M}_{ik}^2] &= \mathbb{E}[\text{Var}(M_{ik} | \mathbf{P}_i)] \\
 &= 2\mathbb{E}[\text{Var}(M_{ik}^h | \mathbf{P}_i)] \quad \because M_{ik}^m \perp\!\!\!\perp M_{ik}^p \text{ (HWE)}
 \end{aligned} \tag{27}$$

Equation 28 at the next subsection finishes the proof. Since we assume the case of two ancestries,

$P_{i1} + P_{i2} = 1$, hence $P_{i1}(1 - P_{i1}) = P_{i2}(1 - P_{i2}) = P_{i1}P_{i2}$, thereby yielding Equation 11

Reviewer: "Dropping the marker variables from the regression gives the interpretation of admixture mapping...". Do you simply mean that the tractor model (Equation S3), without the markers, the same as the model of admixture mapping (Equation S4)? If so, what does it add to the argument?

Response: The initial attempt was to show that the properties of admixture mapping follow as a byproduct of analyzing Tractor. As mentioned earlier, we removed these parts to make the draft clearer.

Reviewer: The supplementary note derives results for three distinct models in parallel. As a result, the reader must infer which equation applies to each model based on subtle distinction between notations (e.g., beta with two subscripts, or bold beta with one subscript is the tractor model, beta with one subscript is Armitage, etc). This would have been a lot easier I think to present them sequentially. I do appreciate that the authors want to draw on the parallels between the models, but then I think that a clearer notation would be helpful.

Response: We extensively restructured the manuscript. Also, we dropped the admixture mapping portion from the manuscript and focused on comparing standard GWAS (ATT) and Tractor. We hope this made an improvement.

Reviewer: Abstract: "can make empirical predictions on GWAS" that is a very vague statement. It would be useful to be explicit about what kind of predictions we are talking about. I.e., variance of parameter estimates.

Response: We changed our abstract to :

Abstract

Admixed populations offer valuable insight into the genetic architecture of complex traits. Many studies have proposed methods for genome-wide association study (GWAS) in admixed populations and various simulation studies have evaluated their performances. In this work, we propose another direction of comparison of recently proposed methods for admixed GWAS from a population genetic viewpoint. Our theoretical approach can mathematically and directly compare the power of methods given that the causal variant is tested. This is done by deriving the variance formula of the methods from the population genetic admixture model. Our results analytically confirm previous observation that the standard GWAS test is more powerful than alternative tests due to leveraging allele frequency heterogeneity in which alternatives do not. As a by-product, we obtain a simple method to improve the power of multi-degrees-of-freedom tests only using summary statistics. We further investigate the problem when the causal variant is not directly known but is detected by tagging variants in linkage disequilibrium (LD). The analysis shows that a genetic segment from admixed genomes may exhibit distinct LD patterns from the single-continental counterpart of the same ancestry.

Reviewer: Figure 3b: Is the overlap significant? Or could the long-range LD be mostly noise? I do see that there is some amount of overlap for very short segments, but it is very hard to say whether this is relevant.

Response: We added the connection of Fig 4B to other parts of the manuscript. Also, we modified the figure to include only the LDs at a shorter distance which is relevant for many downstream methods such as LDSC and fine-mapping (<1cM).

Reviewer: "As we show, ... other methods do not" I don't think that this has been shown.

Response: Yes. This part was omitted in the initial version. We removed the phrase and restructured the manuscript. We agree that it is unclear what "intuitive" means in this context. For some clarification, we added annotations that tell which part of the signal comes from either LD or admixture.

$$\begin{aligned}
\beta_k = & \sum_{j \in [k]} \frac{\mathbb{E}[\sum_l D_{ljk} P_{il}]}{\underbrace{\mathbb{E}[\sum_l g_{lk}(1-g_{lk})P_{il} + \sum_l g_{lk}^2 P_{il}(1-P_{il}) - \sum_{l \neq l'} g_{lk} g_{l'k} P_{il} P_{il'}]}_{\text{Within-continental LD}}} \alpha_j \\
& + \sum_{j=1}^{n_J} \frac{\sum_{l,l'} g_{lk} f_{l'j} \mathbb{E}[\text{Cov}(L_{ikl}^h, L_{ijl'}^h | \mathbf{P}_i)]}{\underbrace{\mathbb{E}[\sum_l g_{lk}(1-g_{lk})P_{il} + \sum_l g_{lk}^2 P_{il}(1-P_{il}) - \sum_{l \neq l'} g_{lk} g_{l'k} P_{il} P_{il'}]}_{\text{Admixture LD}}} \alpha_j
\end{aligned} \tag{5}$$

220

$$\beta_{lk} = \sum_{j \in [k]_l} \frac{D_{ljk}}{g_{lk}(1-g_{lk})} \alpha_j \quad (l = 1, \dots, n_L) \tag{6}$$

Within-continental LD

$$\gamma_{lk} = \sum_{j \in [k]_l} \left(\frac{f_{lj} - h_{lj}}{1-g_{lk}} - \frac{f_{1j} - h_{1j}}{1-g_{1k}} \right) \alpha_j \quad (l = 2, \dots, n_L) \tag{7}$$

Reviewer: The sum of Bernoulli is Binomial only if probabilities are equal, and I don't think they are here in general. Do you mean Poisson Binomial? This error came up a few times. Are you assuming that maternal and paternal alleles have the same ancestry?

Response: No we do not assume that the parental alleles have the same ancestry. The original description was our mistake in language and the actual derivation did not assume that. We now have updated the description to prevent confusion.

141 Local ancestry (LA) is assigned according to the probability specified by the global ancestry. At
142 locus k , local ancestry L_{ikl} counts how many copies of the locus originated from ancestry l . In diploids,

4

including humans, $\sum_{l=1}^{n_L} L_{ikl} = 2$. It follows a multinomial distribution

$$\mathbf{L}_{ik} \mid \mathbf{P}_i \sim \text{Multinomial}(n = 2, p = \mathbf{P}_i) \quad (1)$$

where $\mathbf{L}_{ik} = [L_{ik1}, \dots, L_{ikn_L}]^T$ and $\mathbf{P}_i = [P_{i1}, \dots, P_{in_L}]^T$. Precisely speaking, $\mathbf{L}_{ik}^{\mathbf{h}}$ ($\mathbf{h} = \mathbf{m}$ for
maternal and $\mathbf{h} = \mathbf{p}$ for paternal haplotypes) is sampled from $\mathbf{L}_{ik}^{\mathbf{h}} \mid \mathbf{P}_i \sim \text{Multinomial}(n = 1, p = \mathbf{P}_i)$
and $\mathbf{L}_{ik} = \mathbf{L}_{ik}^{\mathbf{m}} + \mathbf{L}_{ik}^{\mathbf{p}}$. Finally, the genotype $G_{ik}^{\mathbf{h}} = 0, 1$ of haplotype \mathbf{h} is sampled from a Bernoulli
distribution $G_{ik}^{\mathbf{h}} \mid L_{ikl}^{\mathbf{h}} = 1 \sim \text{Bernoulli}(p = f_{lk})$ conditional on $\mathbf{L}_{ik}^{\mathbf{h}}$ where l is the source ancestry of
haplotype h at locus k . f_{lk} is the reference allele frequency at the locus in ancestry l .

Reviewer: First results page, line 7, missing symbol. Ambiguous subject to "summing". L18: allele frequenc*ies*, presumably? L29: "that 'admixture events occurred a few dozens of generations ago:" This is not true of all admixture events! L43-45: variance of what? Of the estimated effect size? Results, p2, L 48: casual - causal

Response: These parts were removed after restructuring the manuscript. The last form of typo has been corrected.

Reviewer: Figure 1A: It may help the non-statistical reader to explain why it is important to get an analytical form for the standard error if we can estimate these errors from the data itself.

Response: We added some details on how admixture reduces the standard error and how that can be shown using the formula.

ATT's (the standard GWAS) test statistics is always larger or equal to the Tractor's statistics. This
advantage is driven by the allele frequency difference $g_{1k} - g_{2k}$ between the two source populations.
This part explains how admixture LD contributes to power. The coefficient being tested remains
the same, but admixture LD increases the test statistics by improving the precision of the estimate,
i.e., a smaller standard error and a narrower confidence interval. The combined statistics $T_{n_l}^{\text{Tractor}}$
additionally suffers from the increased degrees-of-freedom of the null χ^2 -distribution. What's new is
that Tractor is less powerful than standard GWAS even without the effect of increased degrees-of-
freedom because the absolute value of the test statistics is smaller. Also, Tractor does not benefit
from allele frequency heterogeneity across ancestral populations at all, although it still does from
heterogeneous effect sizes. This means that admixed GWAS is unlikely to add more statistical power
than single-continental GWAS if one uses Tractor in the absence of a conspicuous causal effect size
heterogeneity. The gain of ATT from $(g_{1k} - g_{2k})^2 \mathbb{E}[P_{i1}P_{i2}]$ depends on $\mathbb{E}[P_{i1}P_{i2}] = \mathbb{E}[P_{i1}(1 - P_{i1})]$,
which is larger when individuals with equal ancestral proportions from both ancestries are common
in the population (function $f(x) = x(1 - x)$ is maximized at $x = 0.5$). It is important to note that
the power gain driven by $(g_{1k} - g_{2k})^2 \mathbb{E}[P_{i1}P_{i2}]$ is absent in a multi-ancestry cohort only made up of
people of single-continental origins. People from ancestry l will have $P_{il} = 1$ and $P_{il'} = 0$ for $l' \neq l$,
so $\mathbb{E}[P_{il}P_{i2}]$ is always zero.

Reviewer: Figure 1B: Please define acronyms. Assuming FE: Fixed Effects meta-analysis, etc.

Response: The new legend now expands the acronyms.

Figure 2: Predictions of the PSD model evaluated in real data. **A.** Comparison of predicted and estimated standard error of regression coefficients. The top panel is for height and the lower panel is for body-mass index (BMI). **B.** Quantile-Quantile (QQ) plot GWAS results of height (left) and BMI (right) in the PAGE cohort. FE: Fixed-effects meta-analysis, RE2: Han-Eskin random-effects meta-analysis.

Reviewer: Figure 3a: The scale goes from -.5 to -4.5. So independent of the logarithmic basis used (incidentally, which one?) all distances are less than 1 bp. Do you mean Mbp? This is important to understand the scale of the LD inaccuracies

Response: All the scales are now in absolute scale so that they are more readable.

Reviewer #2

Reviewer: This paper presents evidence to support a claim that widely adopted methods for leveraging local ancestry estimates in admixed genomes for GWAS purposes is flawed. Specifically, mathematical theory and results using both simulated and read data are used to show that a mosaic of single-continental segments as an approximation of admixed populations is not sufficient to enable previously accurate estimation of two-locus effects. Some of the results are interesting, however further detail is required to fully support the claims made and to establish a clearer picture of the issue presented. I do think the work is worth pursuing and that a picture is being built of an important issue. However the paper feels incomplete and deserves further work.

Response: We thank the reviewer for carefully reading our paper. We did not intend to make such a strong claim that the method currently being used is flawed. We tried to understand and compare LD patterns of chromosomal segments from the same ancestry in admixed and single-continental genomes, unlike previous works that compared segments from different ancestries in only admixed genomes. We believe that some highly overloaded notation and overlap of authors with previous works have created substantial confusion. We apologize for this mistake and hope that the revised manuscript makes these points clearer. We added an overview figure for this purpose.

Figure 1: Overview of this study. **A.** Power of various GWAS methods applied to different types of cohorts (single-ancestry, multi-ancestry, and admixed) were mathematically compared. **B.** European segments from European genomes were compared to segments of the same ancestry from admixed genomes. The same comparison was made for African segments, too.

Reviewer: Firstly, much of the methodology and analysis is unclear. The mathematical results are relegated to a supporting document which is sparsely and inaccurately referenced. The simulation study is vaguely documented, such that results that depend on the simulation settings cannot be extrapolated. The results provided on the real data feel somewhat cherry-picked and far from adequate.

Response: We restructured and rewrote most parts of the manuscript to prevent tedious back-and-forth between the main section and supplementary materials. We added the following sections to explain the precise simulation setting and software. Selective reporting of real data analysis is mentioned in the other comment. Briefly, we included the results of other traits in the Supplementary Figures 2-7. The following figure is Supplementary Figure 2 and rest of the new figures can be found in the supplementary material.

Supplementary Figure 2. Estimated versus predicted standard error of Tractor and standard GWAS regression coefficients or three quantitative traits. The traits are on the left most of the figure.

Reviewer: The notation and terminology is inconsistent with the literature, including publications by some of the authors of this paper. This leads to difficulty in establishing exactly which methods from the literature are being critiqued. Elements of the figures are not discussed in the text.

Response: We removed heavily overloaded notations (for example, r^2_g). We hope that the restructured draft better aligns with previous literature. We added additional legends and descriptions to the figures. We hope that the new manuscript is sufficiently self-contained for the reader to follow.

Reviewer: Finally, the contribution of this paper as stated is not large. Previous related work already established that concordance of effect estimates across admixture components is good for almost all phenotypes for causal markers. This paper focusses on the lack of such concordance for SNPs not deemed to be causal. These are essentially false-positive or even true-negative SNPs; this paper does not discuss how causal SNPs are identified, but presumably most research that attempts to leverage the ePSD model that is critiqued here are interested only in the causal SNPs. Of course these are not typically known but rather inferred or simply tagged by SNPs in high LD with them, however in that case this (and previous) papers show that the ePSD framework performs well.

For example, "Causal effects on complex traits are similar for common variants across segments of different continental ancestries within admixed individuals" advocates focussing on causal effects. This finding is corroborated in this paper, therefore the manuscript can be summarised as finding that non-causal variants effect sizes are not well preserved across ancestries; this is not surprising as the true effect size is zero. i.e. false positive and true-negative effect sizes are not consistently estimated across ancestries.

Response: The key difference between Hou 2023 paper and this work is that Hou 2023 compares genomic segments from the same admixed individuals from different ancestries, while this paper compares segments from the same ancestry but from different individuals. By doing so, Hou (2023) tried to identify the causal effect size heterogeneity across different ancestry segments, while this paper attempted to identify the LD pattern heterogeneity between the same ancestry segments.

The analysis presented in Hou 2023 uses a new method to compute the genetic correlation of causal effect sizes of different ancestral origins. The method includes all variants, unlike GWAS doing marginal testing. The reason for including all variants are explained in section "Pitfalls of using marginal effect sizes to estimate heterogeneity" and Fig 4 (Hou 2023), which shows different LD patterns in segments of different ancestral origin leads to marginal effect size heterogeneity. Since all variants are included in the model, each variant does not pick up signals from other variants from LD. This allowed the model of Hou 2023 to compare the causal effect without worrying about heterogeneous LD patterns that induce heterogeneity in the marginal effects. As the model does not depend on the LD pattern, the results in the paper have no connection to the validity of ePSD.

This work, on the other hand, shows that the LD patterns, and not the causal effect size, are different in segments of the same ancestral origin. For example, we computed the LD pattern of a variant in admixed genomes that reside on an African segment. Then, we compared this with the LD pattern computed from single-continental African genomes. The same procedure was conducted for European segments.

To repeat, we compared African segments from admixed genomes to African segments from admixed genomes. Hence, unlike the previous work of Hou et al (2023) that compared segments of different ancestries, this work compared segments of the same ancestral origin. Likewise, European segments from admixed genomes were compared to European segments from single-continental genomes. See Figure 1B and Supplementary Figure 1 for a visual comparison.

Figure 1: Overview of this study. **A.** Power of various GWAS methods applied to different types of cohorts (single-ancestry, multi-ancestry, and admixed) were mathematically compared. **B.** European segments from European genomes were compared to segments of the same ancestry from admixed genomes. The same comparison was made for African segments, too.

Supplementary Figure 1. Overview of the analysis of Hou et al. (2023). African and European segments only from admixed genomes were compared to each other.

Reviewer: The commentary paper "Estimation of cross-ancestry genetic correlations within ancestry tracts of admixed samples" by Atkinson summarises the results of the "Causal effects..." paper and states that "The most important finding from this work is that causal effects for common variants were largely similar across ancestries, with height notably bucking the trend and showing a significant admix < 1 ". Again, the key difference between that work and this manuscript is the focus on causal variants, right?

Response: As noted above, we measure the correlation of marginal effects between African segments from admixed and single-continental genomes. We did it, too, for the European segments. Hence, African segments are compared to African segments, and European segments are compared to European segments. After reading one of your comments below, we suspect that our highly overloaded notation caused the confusion. Despite the different goals of our paper and Hou (2023), we used the same notation $r_{g, admix}$ to denote a different quantity.

Reviewer: Thus the key finding that "the concordance dropped with the increasing proportion of causal loci" as per Fig 2(c) is already established in the "Causal effects" paper in Fig 6(d,e,f), where the density of causal effects is increased in simulations. This diminishes the novelty of this paper. Similarly, another previous work "On powerful GWAS in admixed populations" showed that "GWAS in admixed populations attain improved power for discovery over homogeneous populations in either scenario - similar or different ancestry-specific allelic effects - thus further supporting the need for larger genomic studies in such populations." Hence this paper's contribution appears small in comparison.

Response: For the reasons mentioned above, this paper is measuring the concordance between African segments from admixed and single-continental genomes which is comparing segments of the same ancestry (and also for the Europeans). Therefore, it is completely different from comparing African segments to European segments of admixed genomes which compares segments of different ancestry.

Although the power advantage of standard GWAS over Tractor has been repeatedly verified in earlier studies, our analysis explains where the advantage exactly comes from and its precise magnitude in terms of allele frequency parameters. We mathematically show that Tractor does not benefit from heterogeneous allele frequency across populations. Furthermore, we show that admixed individuals confer additional power gain compared to a multi-ancestry cohort consisting of multiple single-continental individuals.

ATT's (the standard GWAS) test statistics is always larger or equal to the Tractor's statistics. This
advantage is driven by the allele frequency difference $g_{1k} - g_{2k}$ between the two source populations.
This part explains how admixture LD contributes to power. The coefficient being tested remains
the same, but admixture LD increases the test statistics by improving the precision of the estimate,
i.e., a smaller standard error and a narrower confidence interval. The combined statistics T_{nl}^{Tractor}
additionally suffers from the increased degrees-of-freedom of the null χ^2 -distribution. What's new is
that Tractor is less powerful than standard GWAS even without the effect of increased degrees-of-
freedom because the absolute value of the test statistics is smaller. Also, Tractor does not benefit
from allele frequency heterogeneity across ancestral populations at all, although it still does from
heterogeneous effect sizes. This means that admixed GWAS is unlikely to add more statistical power
than single-continental GWAS if one uses Tractor in the absence of a conspicuous causal effect size
heterogeneity. The gain of ATT from $(g_{1k} - g_{2k})^2 \mathbb{E}[P_{i1}P_{i2}]$ depends on $\mathbb{E}[P_{i1}P_{i2}] = \mathbb{E}[P_{i1}(1 - P_{i1})]$,
which is larger when individuals with equal ancestral proportions from both ancestries are common
in the population (function $f(x) = x(1 - x)$ is maximized at $x = 0.5$). It is important to note that
the power gain driven by $(g_{1k} - g_{2k})^2 \mathbb{E}[P_{i1}P_{i2}]$ is absent in a multi-ancestry cohort only made up of
people of single-continental origins. People from ancestry l will have $P_{il} = 1$ and $P_{il'} = 0$ for $l' \neq l$,
so $\mathbb{E}[P_{il}P_{i2}]$ is always zero.

Reviewer: Conversely, "On powerful GWAS in admixed populations" states that "Existing association tests attain increased power over traditional GWAS in admixed populations, even when the causal variant has similar allelic effects across ancestries". It also states that "there is an expected loss of power due to imperfect tagging, although preliminary results suggest that the loss in power is small, particularly when genotype imputation is employed". This appears at odds with the results presented in this paper. If so, a direct repudiation should be included, stating how that analysis differs or is incorrect. Or how the scenario considered differs, whichever is relevant.

Response: We clarified how ATT benefits from admixture, unlike Tractor, in response to the previous comment. We believe that we have not claimed on how tagging affects power. In the case of non-causal marker variants, the true marginal effect size differs between methods. Hence, we limited our analysis to the case of testing the causal variant. Overall, our analysis is consistent with previous empirical investigations and adds further theoretical clarification that explain those results.

**2.5 The Pritchard-Stephens-Donnelly model can predict GWAS power**

Standard GWAS (the Armitage trend test, ATT) and Tractor are two popular methods for conducting
GWAS in admixed populations. Simulation-based and empirical comparisons have previously been
made in the literature provided the causal variant was directly tested [44, 45]. We complement the
earlier findings with a precise mathematical formula. In the main text, we present the case of two
source populations. See the **Materials and Methods** for the general result of more than two source
populations. When the tested variant is causal and does not tag any other variants, **Equation 5** and
**6** reduce to

$$\beta_k = \sum_{j=k} \alpha_j = \alpha_k \quad \text{and} \quad \beta_{lk} = \sum_{j=k} \alpha_j = \alpha_k \quad (8)$$

which are identical to the causal effect size of the tested variant.

Reviewer: Unfortunately, some crucial simulation details are either lacking or unsatisfactory. Only chromosome 22 was simulated; this is very short and within a small number of post-admixture generations very few ancestry switches will be observed. Thus the local ancestry inference will be very uncertain, as there are simply too few events to train on. What reference panels are used to perform local ancestry inference, or is this taken from the underlying ground truth? The manuscript states that Wright-Fisher was used for 5 generations simulation and then coalescent, but for how many more generations?

Response: The simulation was poorly described in the original version. We apologize for this. The simulation duration is depicted in Supplementary Figure 8 where the demographic model was adopted from Browning et al. (2018).

Supplementary Figure 8. The demographic model of African-American admixture adopted from Browning et al. (2018). The demes yaml file was retrieved from stdpopsim catalog.
Link: <https://popsim-consortium.github.io/stdpopsim-docs/stable/catalog.html>

We used a longer genome ($\approx 10^8$, roughly the half the size of Chr3), adopting the parameters from chromosome 1. We extracted the local ancestry information from the simulation directly using tskit and tspop, rather than inferring them. Coalescent simulation is retrospective (backwards-in-time), so we cannot fix the number of generations like forward simulations. The simulation lasts until the last coalescence event occurs. In this case, it continues through the pre-out-of-Africa period due to the demographic model where the last coalescent event happens inside Africa. The first five generations of Wright-Fisher refers to turning off the coalescent approximation option in msprime in the first few generations (backwards-in-time) following the suggestion of Nelson et al. (Accounting for long-range correlations in genome-wide simulations of large cohorts).

We added the following:

4.6 African-American genome simulation

The demographic model of African Americans was retrieved from `stdpopsim` 0.2.0 catalog [53]. The
model id we used was `AmericanAdmixture_4B11` that was reported in Browning and colleagues [51].
We excluded the East Asian contribution and modified the admixture proportion of admixed African
Americans to be 80% African and 20% Europeans. We assumed that the admixture occurred 12
generations ago [32, 21, 33]. The resulting demography model file is available in our paper’s github in
`demes` format [66]. A visual presentation of the demographic model is in **Supplementary Figure 8**.

Using the above demography, we simulated 5000, 5000, and 10000 African, European, and African
American individuals with `msprime` 1.3.3 [50, 67]. The genome length was set to 10^8 base pairs
(approximately half the length of chromosome 3). Recombination and mutation rate were set to
1.15×10^{-8} and 1.29×10^{-8} which were adapted from *Homo sapiens* `stdpopsim` catalog [53]. Following
the suggestion from Nelson and colleagues, the first 5 generations (backwards in time) were set to
follow a Wright-Fisher process [57]. For the remaining period in the past, the default coalescent
process of `msprime` was used. Local ancestry information of African Americans were extracted from
the tree sequence generated by `msprime` using `tspop`. By recording all the lineages in the tree sequence

that were present right before the admixture event, `tspop` tracks the ancestry segments of the modern
genome to their ancestors in the ancestral source populations [68].

Note that the number of generations in coalescent simulations is not fixed. It proceeds until all lineages coalesce. Given that the coalescent process is a random stochastic process, the number of generations here is a random variable, rather than a fixed parameter. Under the demographic model in Supplementary Figure 8, the simulation will continue at least until the out-of-Africa event because all lineages should at least the blue bar (African) before them to completely coalesce.

Reviewer: The statement in the abstract that "Our results show that a mosaic of independent single-continental segments is an insufficient approximation of contemporary admixed populations" overstates the results presented; the finding relates only to the concordance of effect size estimates.

Response: The paper shows that given that ePSD is correct, the ancestry-specific estimates of Tractor should be identical to the single-continental counterpart. Since we see a huge discrepancy between the two types of estimates, we provide evidence that ePSD is wrong, which is a proof by contradiction. However, we agree that ePSD cannot be ruled out completely by this single line of evidence.

2.6 Testing the extended PSD model in real data and simulations

As **Equation 6** is deduced from the ePSD model, we can indirectly assess the model by testing
**Equation 6**. The equation says that Tractor's ancestry-specific estimates are identical to the sum-
mary statistics had the GWAS conducted on single-continental genomes. Given that the prediction
is correct, we hypothesized that measuring the genetic correlation of Tractor's African and European
effect sizes with the corresponding single-continental summary statistics will produce genetic corre-
lations close to 1 provided the ePSD is correct. Tractor estimates were obtained from the PAGE
cohort and the single-continental summary statistics were obtained from PanUKBB. We used link-
age disequilibrium score regression (LDSC) to estimate the genetic correlation (see **Materials and**
**Methods**).

Indeed, local ancestry inference methods and, more generally, the coalescent-with-recombination on which those methods are based do not assume that different ancestry blocks are independent. We explain why an independent assumption is frequently adopted in GWAS literature in the following paragraph and beyond:

2.3 The extended PSD and its connection to previous literature

So far, we explained how genotype at a single locus is determined by the PSD model. The model
does not describe the dependence between two or more loci and models the joint distribution as if
the loci are mutually independent [1, 25]. Local ancestry inference methods, whether discriminative
(RFMix) [8] or generative (hidden Markov model-based methods) [7, 10], employ an approximate
coalescent with recombination model to describe the dependence between the loci [25, 26, 27]. In this
framework, LD-related quantities of the ancestral and admixed populations such as LD covariance are
realizations of the aforementioned coalescent process parameterized by the recombination rate and
other evolutionary parameters [28], opposed to most, if not all, studies in GWAS literature that treat
the LD-related quantities as a fixed parameter [29, 30].

Reviewer: A more explicit framing of how this paper relates to others in the literature is therefore required. This paper attempts to set out mathematical findings and results based on both simulated and real data to support a claim that existing methods perform poorly. Figure 2 is central to this; there is very low concordance between effect sizes in African versus European ancestries. However, 2(c) confirms results from the literature that concordance is high for causal variants, especially when there are not too many of them. Again, this reduces the contribution of this paper to saying that variants that are not causal do not have consistently estimated effect sizes. How important is this? Should such variants not simply be discarded once determined they are not causal? If establishing which variants are causal is the issue, this should be made clearer and how causal SNPs are chosen in the results presented (e.g. in Figure 2 (c)) do this.

Response: As stated in the earlier responses, we did not claim that the effect sizes in African versus European ancestries are different. As you've correctly pointed out, this was demonstrated in earlier papers such as Hou 2023. In this work, we compared marginal effect sizes and the LD pattern in African segments from admixed genomes versus African segments from single-continental genomes. We added annotations (highlighted in red) to the figures, as well as descriptions in the manuscript and the methods section.

The local ancestry-adjusted LD covariance in **Equation 13** was computed by regressing C_{ij} on
 M_{ik1} , M_{ik2} , and L_{ik2} in 10000 admixed genomes. The coefficient of M_{ik1} , corresponding to Africans,
 was compared to the coefficient of M_{ik} obtained from 5000 African genomes. Similarly, the coeffi-
 cient of M_{ik2} was compared to the coefficient of M_{ik} obtained from 5000 European genomes. These
 comparisons are presented in **Figure 4A**.

We greatly agree that drawing connections with previous works is essential to position our work in the literature:

2.3 The extended PSD and its connection to previous literature

So far, we explained how genotype at a single locus is determined by the PSD model. The model
does not describe the dependence between two or more loci and models the joint distribution as if
the loci are mutually independent [1, 25]. Local ancestry inference methods, whether discriminative
(RFMix) [8] or generative (hidden Markov model-based methods) [7, 10], employ an approximate
coalescent with recombination model to describe the dependence between the loci [25, 26, 27]. In this
framework, LD-related quantities of the ancestral and admixed populations such as LD covariance are
realizations of the aforementioned coalescent process parameterized by the recombination rate and
other evolutionary parameters [28], opposed to most, if not all, studies in GWAS literature that treat
the LD-related quantities as a fixed parameter [29, 30].

To elaborate, modern admixed genomes are thought of as a realization of the evolutionary process
that began at the time of admixture happened in the past. Coalescent with recombination explicitly
models the recombination process backwards in time [31]. The observed LD patterns in the modern
genome are merely one of the many possibilities that could have materialized from this random process
[28]. Note that current allele frequencies of the source and admixed populations are also realizations
of the random process in this setting due to genetic drift and mutation [7, 25]. In GWAS literature,
we implicitly condition the current state of the population and treat the contemporary LD patterns
as a parameter of the current population [29, 30]. For instance, the famous linkage disequilibrium
score regression (LDSC) uses the sample LD scores estimated from the reference panel to approximate
the population LD correlation of contemporary populations. It does not make any reference to the
underlying evolutionary process of recombination. Only the realizations of the process as a collection
of LD-related parameters are considered.

By changing Figure 4A, we provide a clearer picture at what distance the ePSD model start to fail significantly. The range is between 0.1 to 1cM (or 100kb to 1000kb) which is within a range that is considered local in the literature. For example, LD scores are usually computed in a 1cM window. In this range of window, causal variants cannot be identified without the help of fine-mapping techniques, and statistical fine-mapping requires correct LD information around the lead variant. Therefore, discordance of marginal effect size is a practically important issue.

A

We repeatedly highlight that the x-axis is the LD covariance in African (or European) segments computed from admixed genomes and the y-axis is the LD covariance in African (or European) segments computed from single-continental genomes. African LDs are compared to each other and European LD are compared to each other. We never compared African LD to European LD in this work. See Figure 1B and Supplementary Figure 1 for a visual comparison of the two studies.

Reviewer: The claim that "comparing LD correlations in admixed and single-continental genomes revealed only moderately concordant patterns" does not seem fully justified. Far apart loci (yellow in Figure 3(a)) have differing levels of LD in admixed versus single-continental genomes. How much is this due to the simulation settings? It's impossible to tell as the specifics of the simulation are not provided. Were both sets of genomes simulated for the same number of generations and population size?

Response: We apologize for the poor description of the simulation settings. The method section now contains the details. Briefly, we split the window size to make the figure more readable and discarded too distant pairs of variants (Fig 4A). The simulation has been conducted as mentioned in the earlier responses. All the individuals were collected from the same run of simulation that follows a demographic model of modern American population starting from Africa in the past (Supplementary Figure 8).

Reviewer: The result that there is a substantial overlap in the distribution of lengths of local ancestries and local ancestry adjusted LD correlations provided in Figure 3(b) is compelling and does lead to concern that inclusion of local ancestry is not sufficient to achieve the aims of the

ePSD approach. Perhaps this work can make an important contribution to the literature, but greater care is required to build the case.

Response: We have written the following section to draw connection between LDSC analysis and the subsequent simulation analyses.

Indeed, comparing normalized LD covariances (**Equation 13 and 14**) of admixed and single-
continental genomes revealed only moderately concordant patterns. Here, we compared African
(European) segments from admixed individuals to the corresponding segments in single-continental
(non-admixed) African (European) genomes, which is different from a previous study that compared
segments of different ancestries (African versus Europeans) all obtained from admixed individuals
24. We binned the pairs of loci according to their physical distance (**Figure 4A**). The correlation
was fairly high within the range ($\sim 0.1\text{cM}$ or 100kb) of within-continental LD blocks 54 55, but con-
siderably low in the ranges ($\sim 1\text{cM}$ or 1000kb) considered in LD scores 29. The correlation's decay
were faster in European ancestry that had lower occupancy in the genome (African:European = 8 : 2).
This is likely because the LA segments from the minority population is more easily surrounded by the
majority LA segments.

Reviewer: What are FE and RE2 in Figure 1(b)? Presumably Fixed and Random Effects models, referred to the single-line paragraph on Meta Analysis. But no details are provided in the text or figure caption.

Response: We added the following descriptions:

The ancestry-specific test statistics of Tractor $T_{n_I}^{l, \text{Tractor}}$ are mutually independent for all pairs
of $l = 1, \dots, n_L$ (**Equation 11** and **Materials and Methods**). This is surprising because all
the ancestry-specific statistics are obtained from the same regression and data. The independence
allows us to combine them using existing meta-analysis methods that combine independent summary
statistics. The widely-adopted fixed-effects (FE) meta-analysis produces a test statistics equal to
the Tractor’s combined test [46, 47]. Despite having the same test statistics value, it only has one
degrees-of-freedom because it tests a different hypothesis. FE meta-analysis tests a single-parameter
hypothesis ($H_0 : \beta_k = 0$) assuming that all ancestry-specific effect sizes are equal, and Tractor tests
a n_L -parameter hypothesis that allows all effect sizes to vary ($H_0 : \beta_{lk} = 0$ for all $l = 1, \dots, n_L$).

10

We applied another meta-analysis based on the Han-Eskin model, commonly abbreviated as RE2
(random-effects two) [48, 49]. When applied to the same PAGE data of height and BMI, we found
that FE and RE2 achieve better power than Tractor’s combined test (**Figure 2B**). RE2 has a 1.5
degrees-of-freedom and is known to be powerful when the effect size estimates are heterogeneous.
The fact that it performs worse than ATT and similarly to FE suggests that the marginal effect size
heterogeneity is not large in the two traits. The results remain the same in other quantitative traits
as well (**Supplementary Figure 6-7**).

Figure 2: Predictions of the PSD model evaluated in real data. **A.** Comparison of predicted and estimated standard error of regression coefficients. The top panel is for height and the lower panel is for body-mass index (BMI). **B.** Quantile-Quantile (QQ) plot GWAS results of height (left) and BMI (right) in the PAGE cohort. FE: Fixed-effects meta-analysis, RE2: Han-Eskin random-effects meta-analysis.

Additional details can be found in the method section.

Reviewer: Why were height and BMI selected from the PAGE study for additional attention? Given that height was listed as the only (of 38) phenotypes tested in "Causal effects..." Hou et al to have an r_{admix} statistically significantly lower than 1 and that it is a component of BMI, this feels selective. Were these the most interesting or most extreme of the 19 traits considered?

Response: The new supplementary figures 2-8 now contains other 12 traits that were available in the PAGE cohort. The initial choice was simply because height and BMI were those with the largest sample sizes (non-missing measurements).

Reviewer: Some methodology is presented under Results, namely the PSD and ePSD models. The latter is poorly explained, in fact I originally thought this referred to the Falush et al 2003 extension of the STRUCTURE model, rather than one in which more accurate estimation of local ancestry, leveraging LD patterns, is applied.

Response: Thank you for pointing out an important reference. The following paragraph and the subsequent ones now include how ePSD and other models are related.

2.3 The extended PSD and its connection to previous literature

So far, we explained how genotype at a single locus is determined by the PSD model. The model
does not describe the dependence between two or more loci and models the joint distribution as if
the loci are mutually independent [1, 25]. Local ancestry inference methods, whether discriminative
(RFMix) [8] or generative (hidden Markov model-based methods) [7, 10], employ an approximate
coalescent with recombination model to describe the dependence between the loci [25, 26, 27]. In this
framework, LD-related quantities of the ancestral and admixed populations such as LD covariance are
realizations of the aforementioned coalescent process parameterized by the recombination rate and
other evolutionary parameters [28], opposed to most, if not all, studies in GWAS literature that treat
the LD-related quantities as a fixed parameter [29, 30].

To elaborate, modern admixed genomes are thought of as a realization of the evolutionary process
that began at the time of admixture happened in the past. Coalescent with recombination explicitly
models the recombination process backwards in time [31]. The observed LD patterns in the modern
genome are merely one of the many possibilities that could have materialized from this random process
[28]. Note that current allele frequencies of the source and admixed populations are also realizations
of the random process in this setting due to genetic drift and mutation [7, 25]. In GWAS literature,
we implicitly condition the current state of the population and treat the contemporary LD patterns
as a parameter of the current population [29, 30]. For instance, the famous linkage disequilibrium
score regression (LDSC) uses the sample LD scores estimated from the reference panel to approximate
the population LD correlation of contemporary populations. It does not make any reference to the
underlying evolutionary process of recombination. Only the realizations of the process as a collection
of LD-related parameters are considered.

Reviewer: Check the references to Supplementary Equations:

Response: We have rewritten the supplementary note and believe that the references are now correct.

Reviewer: Does 18 show that "variances are inversely proportional to the ancestry-specific marker variances similar to standard GWAS applied to non-admixed genomes" in Page 6 Line 54?

Response: It must be seen in conjunction with equation 17 to get the conclusion. In the new version, see equation 9 and 20.

Reviewer: "Fortunately, a simple analytic expression is deduced for global ancestry adjustment under the PSD model (equation 19 of Supplementary Note)". Is this the correct reference?

Response: It is the correct reference, but we admit that it looks very obscure. In the new version, please see equation 9.

Reviewer: Page 7 Line 3 states that "This method showed improved power over the original Tractor statistics across various quantitative traits (Figure 1b)", however this does not appear to be the case. TRACTOR in orange appears far closer to the 0,1 line. Again, are the "various quantitative traits" here just height and BMI or all 19 mentioned previously?

Response: A method is less powerful if the points are closer to the main diagonal. You can see that the blue line (ATT=Standard GWAS) has smaller P-values, followed by meta-analysis approaches. The original Tractor test is the least powerful. We included all the remaining traits in PAGE cohort in the supplementary figures. All traits follow the same pattern found in height and BMI.

Reviewer: "Causal effects..." by Hou et al uses RFMix for local ancestry deconvolution. Is that the method used in this paper for the real data? For the simulated data, is the ground truth used? If not, how is it estimated? What are the panels used in RFMix?

Response:

Pre-GWAS steps, including local ancestry inference using RFMix, of the real genotype data were identical to the Hou et al. study. We added the following paragraph:

4.1 PAGE and UK Biobank summary statistics

We analyzed summary statistics of one dataset of African-European admixed individuals. Population
Architecture through Genomics and Environment (PAGE) study included 17,299 genotyped individ-
uals with African-European admixed ancestries determined by estimated admixture proportion and
with approximately 6.9 million variants. Detailed steps of quality control and processing, including
GWAS using `admix-kit` and local ancestry inference using RFMix, can be found in **Genotype data**
**processing** section of Hou and colleagues [21]. Summary statistics of 15 traits of UK Biobank par-
ticipants of African and European ancestry were downloaded from the Pan UK Biobank repository
(<https://pan.ukbb.broadinstitute.org/>).

For simulated data, the local ancestry tracts were ground-truth. The software and algorithms are described in the method section.

4.6 African-American genome simulation

The demographic model of African Americans was retrieved from `stdpopsim` 0.2.0 catalog [53]. The
model id we used was `AmericanAdmixture_4B11` that was reported in Browning and colleagues [51].
We excluded the East Asian contribution and modified the admixture proportion of admixed African
Americans to be 80% African and 20% Europeans. We assumed that the admixture occurred 12
generations ago [32, 21, 33]. The resulting demography model file is available in our paper's github in
`demes` format [66]. A visual presentation of the demographic model is in **Supplementary Figure 8**.

Using the above demography, we simulated 5000, 5000, and 10000 African, European, and African
American individuals with `msprime` 1.3.3 [50, 67]. The genome length was set to 10^8 base pairs
(approximately half the length of chromosome 3). Recombination and mutation rate were set to
1.15×10^{-8} and 1.29×10^{-8} which were adapted from *Homo sapiens* `stdpopsim` catalog [53]. Following
the suggestion from Nelson and colleagues, the first 5 generations (backwards in time) were set to
follow a Wright-Fisher process [57]. For the remaining period in the past, the default coalescent
process of `msprime` was used. Local ancestry information of African Americans were extracted from
the tree sequence generated by `msprime` using `tspop`. By recording all the lineages in the tree sequence

that were present right before the admixture event, `tspop` tracks the ancestry segments of the modern
genome to their ancestors in the ancestral source populations [68].

Reviewer: There is inconsistent terminology and notation with closely related works e.g. $R^2 = r_{\text{admix}}$, ePSD is TRACTOR. How does ePSD relate to the methods presented in "Admix-Kit: an integrated toolkit and pipeline for genetic analyses of admixed populations", which is also written by a team including Hou and Pasaniuc?

Response: Admix-Kit uses HAPGEN2 and Haptools2 which are based on the Li-Stephens model. Li-Stephens model is an approximation of the coalescent-with-recombination process that fully describes the neutral evolution of the genome under recombination and genetic drift. These models explicitly model the recombination process. Roughly speaking, ePSD is an even coarse approximation of them that does not explicitly take the recombination process into account.

We elaborate on this point in the following and the subsequent paragraphs:

**2.3 The extended PSD and its connection to previous literature**

So far, we explained how genotype at a single locus is determined by the PSD model. The model
does not describe the dependence between two or more loci and models the joint distribution as if
the loci are mutually independent [1, 25]. Local ancestry inference methods, whether discriminative
(RFMix) [8] or generative (hidden Markov model-based methods) [7, 10], employ an approximate
coalescent with recombination model to describe the dependence between the loci [25, 26, 27]. In this
framework, LD-related quantities of the ancestral and admixed populations such as LD covariance are
realizations of the aforementioned coalescent process parameterized by the recombination rate and
other evolutionary parameters [28], opposed to most, if not all, studies in GWAS literature that treat
the LD-related quantities as a fixed parameter [29, 30].

To elaborate, modern admixed genomes are thought of as a realization of the evolutionary process
that began at the time of admixture happened in the past. Coalescent with recombination explicitly
models the recombination process backwards in time [31]. The observed LD patterns in the modern
genome are merely one of the many possibilities that could have materialized from this random process
[28]. Note that current allele frequencies of the source and admixed populations are also realizations
of the random process in this setting due to genetic drift and mutation [7, 25]. In GWAS literature,
we implicitly condition the current state of the population and treat the contemporary LD patterns
as a parameter of the current population [29, 30]. For instance, the famous linkage disequilibrium
score regression (LDSC) uses the sample LD scores estimated from the reference panel to approximate
the population LD correlation of contemporary populations. It does not make any reference to the
underlying evolutionary process of recombination. Only the realizations of the process as a collection
of LD-related parameters are considered.